# Reprogramming of the transcriptome after heat stress mediates heat hormesis in *Caenorhabditis elegans*

Fan Xu[1,2], Ruoyao Li[1], Erika D. von Gromoff[1], Friedel Drepper [3], Bettina Knapp[3], Bettina Warscheid [3,4,5], Ralf Baumeister[1,2,4,6] & Wenjing Qi [1] ✉

Transient stress experiences not only trigger acute stress responses, but can also have long-lasting effects on cellular functions. In *Caenorhabditis elegans*, a brief exposure to heat shock during early adulthood extends lifespan and improves stress resistance, a phenomenon known as heat hormesis. Here, we investigated the prolonged effect of hormetic heat stress on the transcriptome of worms and found that the canonical heat shock response is followed by a profound transcriptional reprogramming in the post-stress period. This reprogramming relies on the endoribonuclease ENDU-2 but not the heat shock factor 1. ENDU-2 co-localizes with chromatin and interacts with RNA polymerase II, enabling specific regulation of transcription after the stress period. Failure to activate the post-stress response does not affect the resistance of animals to heat shock but eliminates the beneficial effects of hormetic heat stress. In summary, our work discovers that the RNA-binding protein ENDU-2 mediates the long-term impacts of transient heat stress via reprogramming transcriptome after stress exposure.

Heat shock (HS) response is the most studied cellular response to elevated temperatures[1]. It involves rapid transcriptional activation of heat-inducible genes, most notably molecular chaperones like heat shock proteins (HSPs) to detect, refold or degrade unfolded and misfolded proteins, thus preventing their accumulation. HS can alter cellular function in a dose-dependent manner. While prolonged exposure to HS is detrimental, short exposure to HS can even improve stress resistance and extend lifespan, a phenomenon termed heat hormesis[2,3]. For instance, in humans, regular sauna visits have been linked to health-promoting effects attributed to heat hormesis[4]. Studies conducted on model organisms have shed light on the mechanisms underlying heat hormesis and have highlighted the crucial role of Heat-Shock Factor 1 (HSF-1) in activating the transcription of HSPs and

other stress-protective pathways such as autophagy[5,6]. These studies emphasize the importance of improved proteostasis in response to hormetic HS. Up to date, heat hormesis is considered to be a long-term consequence of transcriptional reprogramming during HS exposure, although HS response is characterized by its rapid inactivation upon the termination of the stressor.

ENDU-2 is a member of a family of endoribonucleases called poly-U specific endoribonucleases (ENDOU) that are conserved across many species, including viruses and humans. ENDOU does not share any significant homology with other known ribonucleases, suggesting that it may have a unique function in RNA biology. Although human ENDOU has been proposed in the regulation of tumor development due to its elevated expression level in diverse malignant tumors[7,8], not

[1]Bioinformatics and Molecular Genetics (Faculty of Biology), Albert-Ludwigs-University Freiburg, Freiburg 79104, Germany. [2]Spemann Graduate School of Biology and Medicine (SGBM), Albert-Ludwigs-University Freiburg, Freiburg 79104, Germany. [3]Biochemistry-Functional Proteomics, Institute of Biology II, Faculty of Biology, Albert-Ludwigs-University Freiburg, Freiburg 79104, Germany. [4]Signalling Research Centers BIOSS and CIBSS, Albert-Ludwigs-University Freiburg, Freiburg 79104, Germany. [5]Biochemistry II, Theodor Boveri-Institute, Biocenter, University of Würzburg, 97074 Würzburg, Germany. [6]Center for Biochemistry and Molecular Cell Research (Faculty of Medicine), Albert-Ludwigs-University Freiburg, Freiburg 79104, Germany. ✉e-mail: wenjing.qi@biologie.uni-freiburg.de

much is known about its molecular details and biological roles. Currently, most of the molecular information regarding ENDOU comes from studies in *Xenopus* and viruses. In particular, the presence of an ENDOU homolog Nsp15 in all coronaviruses has drawn significant attention since the outbreak of the coronavirus pandemic. These works together suggested ENDOU to be RNA-binding proteins that cleave poly-U sequences of RNA to affect RNA biogenesis or stability. For example, virus Nsp15 has been found to cleave viral RNA to evade detection by the host defence system[9] and the *Xenopus* homolog XendoU processes the intron-encoded box C/D U16 small nucleolar RNA (snoRNA) from its host pre-mRNA[10]. In addition, both XendoU and human ENDOU have been reported to activate mRNA decay on the membrane of the endoplasmic reticulum (ER) upon $Ca^{2+}$ release, resulting ER remodeling and the formation of tubular ER[11]. Moreover, recent studies in different model organisms have uncovered the conserved function of ENDOU in stress response. In *C. elegans*, ENDU-2 affects animal physiology in response to adverse temperatures, alterations in nucleotide levels, and genotoxic stresses[12,13], while in zebrafish and humans, ENDOU has been shown to cleave and remove the inhibitory 5' uORF of CHOP mRNA, thereby activating CHOP translation upon ER stress[14]. Furthermore, our recent study has shown that ENDU-2 is a secreted protein and can be taken up by the germline to maintain germline immortality upon elevated temperature[15]. Additionally, the presence of signal peptides for ER targeting and secretion in all eukaryotic ENDOU family members suggests that they may play important roles in intercellular communication within multicellular organisms.

In addition to its role in intercellular cross-talk, ENDU-2 also has cell-autonomous functions. Studies have shown that intestinal-expressed ENDU-2 represses the PKA pathway and histone deacetylase HDA-1 in the same tissue, leading to altered mitotic chromosomal segregation in response to nucleotide imbalance and genotoxic stress[12]. ENDU-2 also utilizes its RNA-cleavage activity to decrease the abundance of some of its associated mRNAs, thereby affecting metabolism in the intestine[15]. Thus, ENDOU may combine its cell-autonomous and non-autonomous activities to coordinate stress responses at the organismal level.

In this study, we discover a transcriptional reprogramming after hormetic HS that differs significantly from the acute HS response in *C. elegans*. In addition, the endoribonuclease ENDU-2 controls transcriptional alteration specifically in the post-stress phase but not during HS, thereby mediating the beneficial effects of hormetic HS. We further show that ENDU-2 is associated with chromatin and binds to the promoters of its target genes upon HS. In this way, ENDU-2 promotes the recruitment of RNA polymerase II to affect transcription after HS. Taken together, our work identifies a post-stress response at the transcriptional level and its regulation by ENDU-2 that may have a long-term influence on animals' physiology.

## Results

### ENDU-2 contributes to hormetic heat stress-mediated beneficial effects

ENDU-2 endoribonuclease can be secreted from the soma and taken up into the germline to ensure germline immortality[15]. ENDU-2 also tunes down mRNA levels in the soma in a catalysis-dependent manner at elevated temperatures[15]. Therefore, we speculate that somatic ENDU-2 might have an autonomous protective role upon HS. To test this, we examined the survival of *endu-2(tm4977)* loss-of-function mutant animals at 35 °C and found that loss of *endu-2* did not impair survival of day one adult animals upon persistent HS (Fig. 1a), arguing against ENDU-2 as a protective factor under HS. Although prolonged exposure to HS is detrimental for animals, a short period of HS treatment in early adulthood extends lifespan, a phenomenon known as heat hormesis[16,17]. To test whether ENDU-2 can mediate hormetic HS-induced lifespan extension, we stressed wild-type (WT) and *endu-2(tm4977)* day one adult animals at 35 °C for 1 h and compared their lifespan with that of untreated animals. Corroborating previous reports, 1 h HS increased the lifespan of WT animals (Fig. 1b, Supplementary Fig. 1a). An additional 1 h HS on day 3 of adulthood prolonged lifespan even further (Fig. 1b, Supplementary Fig. 1b). In contrast, lack of *endu-2* completely abolished the beneficial effect of hormetic HS on day one adulthood, and even slightly shortened lifespan after a second HS on day 3 of adulthood (Fig. 1b and Supplementary Fig. 1a, b). In addition, we obtained similar lifespan results with another *endu-2(by188)* loss-of-function allele (Supplementary Fig. 1c). FUDR used in the lifespan assay did not interfere with the heat hormesis, since performing the lifespan in the absence of FUDR treatment yielded similar results (Supplementary Fig. 1d). Hormetic HS has been shown to improve resistance of animals to a subsequent HS and $Cd^{2+}$ toxicity by activating heat shock protein-encoding genes[2,3]. We found that exposing animals to 1 h HS resulted in a significant increase in survival rate of WT but not of *endu-2(tm4977)* animals when they encountered a subsequent HS or $CdCl_2$ treatment (Fig. 1c, d). Moreover, transgenic expression of *endu-2::EGFP* under control of its own promoter completely restored the hormetic HS-induced thermotolerance and partially rescued the lifespan extension defect (Fig. 1d, f). We further explored the reason why our *endu-2::EGFP* transgene failed to completely rescue the lifespan upon hormetic HS. As hormetic HS still significantly extended lifespan of our EGFP knock-in strain *endu-2(by190[endu-2::EGFP])* (Supplementary Fig. 1e), EGFP fusion is unlikely to interfere with the ENDU-2 function. We noticed that our rescue strain expressing *endu-2p::endu-2::EGFP* (BR7295, line 1) had much higher ENDU-2::EGFP protein level than the *endu-2(by190[endu-2::EGFP]* animals (Fig. 1e), we asked whether excessive overexpression of *endu-2::EGFP* might be the cause. We generated two additional *endu-2p::endu-2::EGFP* rescue lines (BR9215, line 2 and BR9217, line 3) via injecting less plasmid so that these two lines had less strong overexpression of ENDU-2::EGFP protein (Fig. 1e). Hormetic HS resulted in a more significantly extended lifespan in these two lines than BR7295 (Fig. 1f and Supplementary Fig. 2a–d). These observations together indicate, although ENDU-2 mediates the beneficial effect of hormetic HS, its excessive overexpression might also have deleterious consequence at the later stage of animals' life that might eventually quench the benefits acquired at the earlier stage.

ENDU-2 has been shown to be expressed primarily in the intestine, in addition to weak expression in some neuronal, muscular and somatic gonadal cells[15]. Secreted ENDU-2 can reach additional tissues. To examine whether the tissue origin of *endu-2* is essential for its involvement in heat hormesis, we expressed transgenic *endu-2::EGFP* in individual tissues of *endu-2(tm4977)* animals and measured hormetic HS-induced lifespan extension and thermotolerance. Intestinal expression of *endu-2::EGFP* (*vha-6* promoter) partially rescued the defective lifespan phenotype in *endu-2(tm4977)* mutant animals upon hormetic heat stress (Supplementary Fig. 2e) and completely restored thermotolerance to WT levels (Fig. 1d). *endu-2::EGFP* expression in muscles (*myo-3* promoter) resulted in a moderate rescue effect in both thermotolerance and lifespan extension (Fig. 1d and Supplementary Fig. 2f). In contrast, neuronal (*unc-119* promoter) or somatic gonadal (*fos-1* promoter) expression of *endu-2::EGFP* failed to rescued both phenotypic aspects (Fig. 1d and Supplementary Fig. 2g, h). It is worth noting that the *myo-3* promoter drives strong expression in all body wall muscle cells, while expression of endogenous ENDU-2 in musculature is rather weak and limited to the head region and anal depressing muscles[15]. These observations together suggest that probably the intestine is the most important tissue expressing *endu-2* to mediate heat hormesis, and muscular expression may play an inferior role.

### Hormetic heat stress results in an ENDU-2-dependent reprogramming of the transcriptome after HS

We previously showed that ENDU-2 acts as a mRNA-binding protein to regulate mRNA stability. Therefore, ENDU-2 may control gene

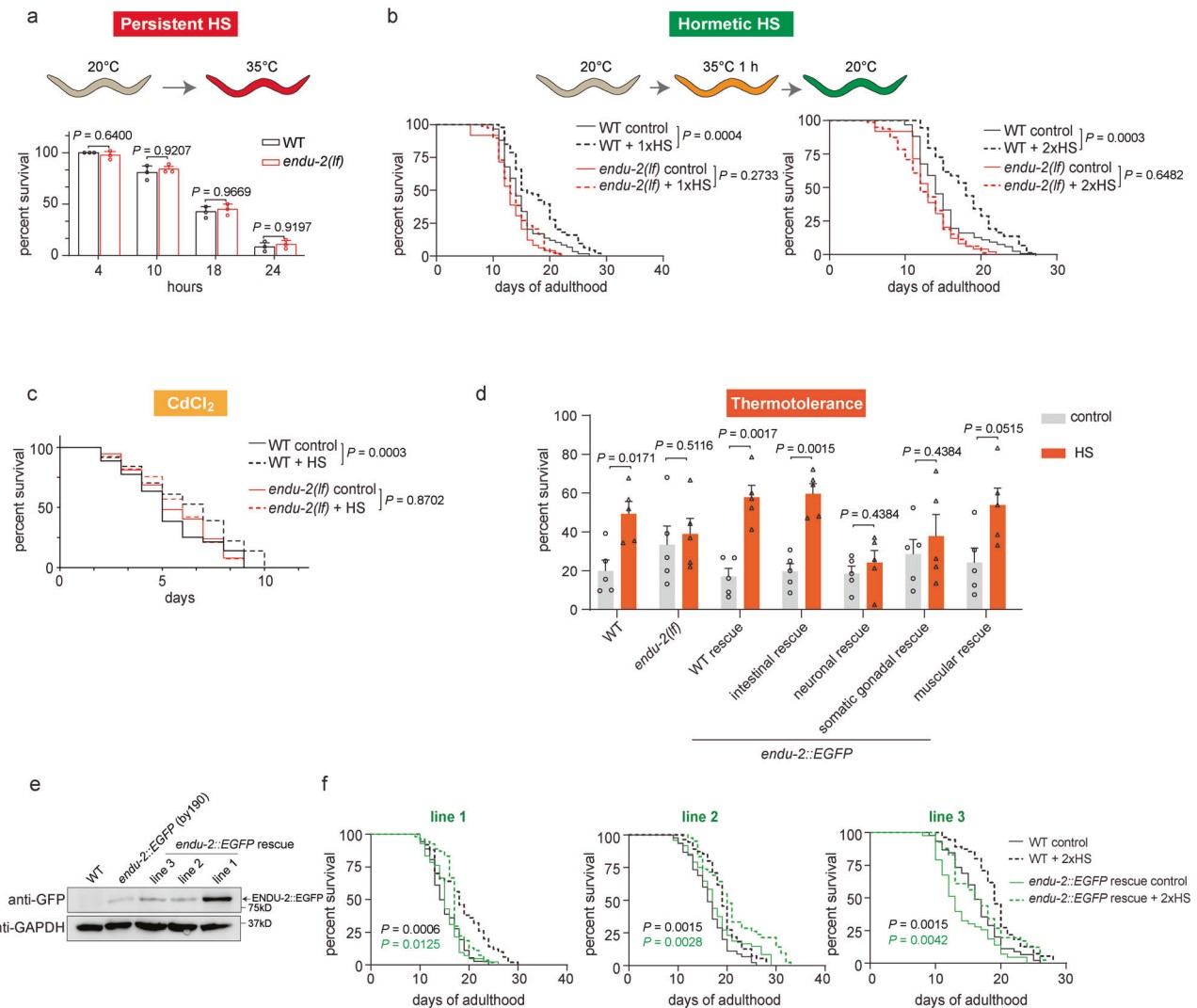

**Fig. 1 | ENDU-2 contributes to hormetic heat stress-mediated beneficial effects.** **a** Survival of wild-type (WT) or *endu-2(tm4977)* animals upon persistent heat shock (HS) for 4 to 24 h at 35 °C. Animals on day 1 adulthood were continuously incubated at 35 °C and scored after 4 h, 10 h, 18 h, and 24 h of HS exposure. Data are the mean ± SD. WT: *n* = 119. *endu-2(tm4977)*: *n* = 111. *N* = 3 independent experiments. *P* values were calculated using two-way ANOVA with Sidak's multiple comparisons. **b** Short exposure to HS in early adulthood extends the lifespan of WT but not *endu-2(tm4977)* animals. Mean lifespan for each group: WT control: 15.1 days, *n* = 59. WT with 1× HS: 18.0 days, *n* = 94. WT with 2× HS: 19.3 days, *n* = 73. *endu-2(tm4977)* control: 13.7 days, *n* = 49. *endu-2(tm4977)* with 1× HS: 13.7 days, *n* = 78. *endu-2(tm4977)* with 2× HS: 12.3 days, *n* = 79. Shown are data from one representative replicate from *N* = 3 independent experiments. The results of all biological replicates are summarized in Supplementary Fig. 1a. *P* values were calculated using Log-rank (Mantel–Cox) test. **c** Survival of WT and *endu-2(tm4977)* animals exposed to CdCl₂ treatment. Day 1 adulthood animals were subjected to 1 h HS at 35 °C followed by 4 h recovery at 20 °C, then animals were exposed to 0.25 mM CdCl₂. Mean survival for each group: WT control: 5.3 days, *n* = 107. WT with HS: 6.4 days, *n* = 108. *endu-2(tm4977)* control: 5.7 days, *n* = 112. *endu-2(tm4977)* with HS: 5.8 days, *n* = 95. Shown are pooled data from *N* = 3 independent experiments. *P* values were calculated using Log-rank (Mantel–Cox) test. **d** Survival of WT, *endu-2(tm4977)*, *endu-2::EGFP* rescued animals after exposure to 35 °C for 8 h. For thermotolerance after hormetic HS, day one adult animals were incubated at 35 °C for 1 h, followed by a 12 h recovery at 20 °C before exposure to the subsequent 8 h HS. Survival rates for each group: WT control: (19.9 ± 6.2) %, *n* = 142. WT with HS: (49.2 ± 5.4) %, *n* = 151. *endu-2 (tm4977)* control: (33.5 ± 8.7) %, *n* = 139. *endu-2 (tm4977)* with HS: (38.9 ± 6.5) %, *n* = 144. *endu-2(tm4977);byEx1375[endu-2p::endu-2::EGFP]* (WT *endu-2*

rescue) control: (17.0 ± 4.2) %, *n* = 133. WT *endu-2* rescue with HS: (57.9 ± 4.5) %, *n* = 139. *endu-2(tm4977);byEx1551[vha-6p::endu-2::EGFP]* (intestinal *endu-2* rescue) control: (19.7 ± 3.8) %, *n* = 150. intestinal *endu-2* rescue with HS: (59.7 ± 4.4) %, *n* = 153. *endu-2(tm4977);byEx1795[unc-119p::endu-2::EGFP::3xFlag]* (neuronal *endu-2* rescue) control: (18.6 ± 3.9) %, *n* = 140. neuronal *endu-2* rescue with HS: (24.3 ± 4.5) %, *n* = 144. *endu-2(tm4977);byEx1379[fos-1p::endu-2::EGFP::3xFlag]* (somatic gonadal *endu-2* rescue) control: (28.6 ± 6.3) %, *n* = 131. somatic gonadal *endu-2* rescue with HS: (37.9 ± 10.2) %, *n* = 133. *endu-2(tm4977);byEx1816[myo-3p::endu-2::EGFP::3xFlag]* (muscular *endu-2* rescue) control: (24.2 ± 6.2) %, *n* = 147. muscular *endu-2* rescue with HS: (53.8 ± 7.1) %, *n* = 142. Data are the mean ± SEM of *N* = 5 independent biological replicates, *P* values were calculated using a two-tailed multiple unpaired *t* test. **e** Western blot detection of ENDU-2::EGFP in WT, *endu-2(by190[endu-2::EGFP])*, *endu-2(tm4977);byEx1375[endu-2p::endu-2::EGFP]* (line 1), *endu-2(tm4977);byEx1908[endu-2p::endu-2::EGFP]* (line 2) and *endu-2(tm4977);byEx1910[-endu-2p::endu-2::EGFP]* (line 3) animals. N2 (WT) animals served as the negative control for ENDU-2::EGFP detection. GAPDH served as an internal control. *N* = 2 independent experiments. **f** Lifespan analysis of WT and *endu-2(tm4977)* animals with *endu-2::EGFP* transgene (lines 1 to 3 are mentioned in **e** upon 2x hormetic HS. Mean lifespan for each group: WT (together with line 1): 15.2 days, *n* = 39. WT (together with line 1) with HS: 18.4 days, *n* = 54. line 1 control: 14.7 days, *n* = 39. line 1 with HS: 17.1 days, *n* = 54. WT (together with line 2 and 3) control: 16.4 days, *n* = 45. WT (together with line 2 and 3) with HS: 19.2 days, *n* = 55. line 2 control: 17.9 days, *n* = 48. line 2 with HS: 21.2 days, *n* = 42. line 3 control: 14.8 days, *n* = 45. line 3 with HS: 18.1 days, *n* = 44. Shown are one representative data of *N* = 3 independent experiments. The results of all biological replicates are summarized in Supplementary Fig. 2a–d. *P* values were calculated using Log-rank (Mantel–Cox) test.

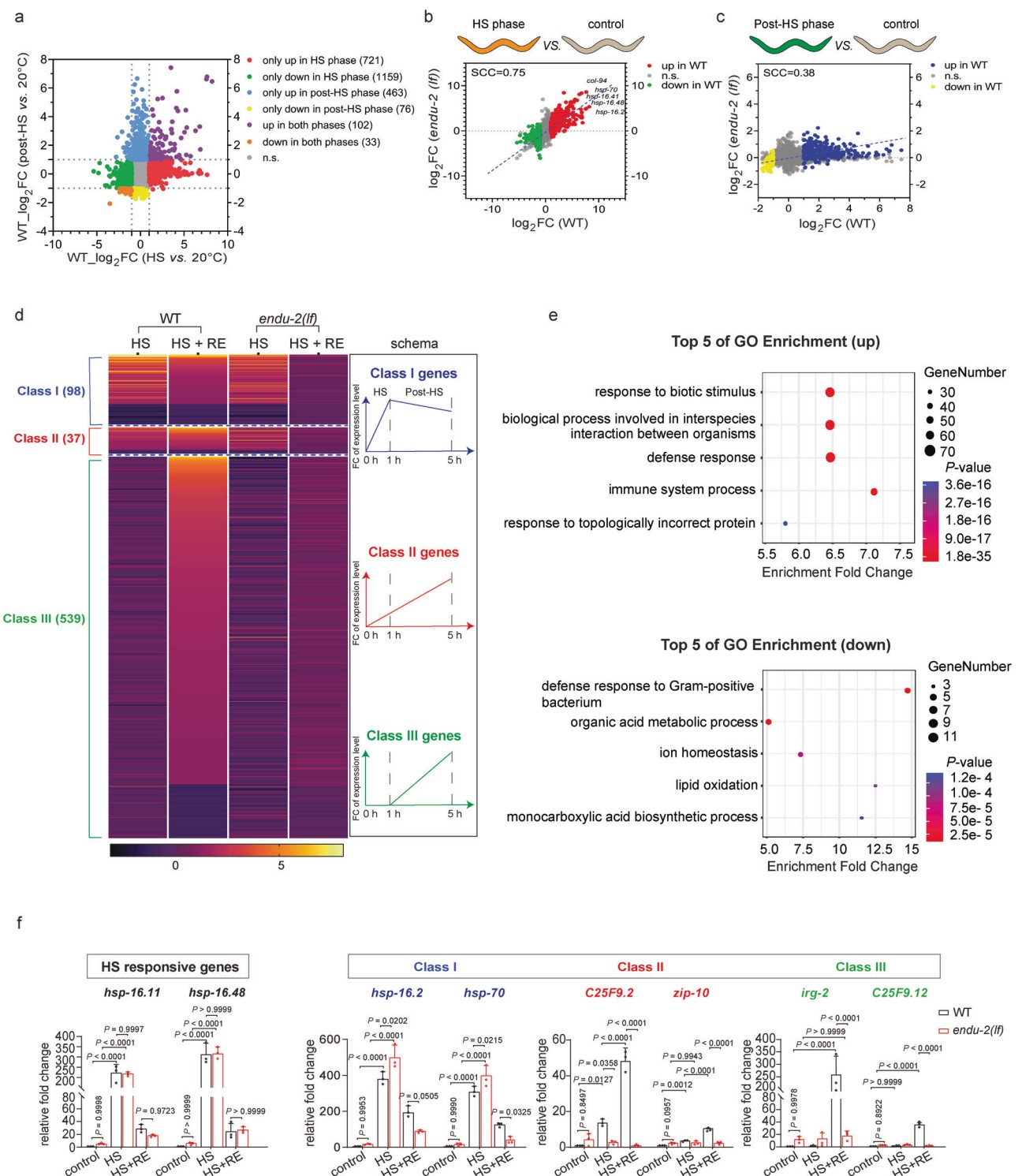

expression at post-transcriptional level to mediate heat hormesis. To know which transcripts are affected by hormetic HS in an ENDU-2-dependent manner, we compared transcriptomes of WT and *endu-2(tm4977)* mutant animals using RNA-seq. As the beneficial effect of hormetic HS is a long-term consequence of a short exposure to stress, we considered that alteration of the transcriptome in the post-stress period might also be relevant. Therefore, we isolated RNA from stage-matched animals that were unstressed, directly after 1 h HS and 4 h recovered after HS (sample preparation and collection illustrated in Supplementary Fig. 3a). The reason for choosing 4 h after HS was to avoid potential interference of internal hatching of larvae that

occurred in a substantial portion of *endu-2(tm4977)* animals at a later stage, as lack of *endu-2* causes severe vulva development defects that leads to animal's death on day 2 and day 3 adult animals[15]. 1 h HS in WT caused an immediate increase of 823 transcripts and decrease of 1192 genes (fold change (FC) > 2, FDR < 0.01) (Fig. 2a, b and Supplementary Data 1). We named these differentially expressed genes (DEGs) during HS "HS responsive genes". The pool of the activated genes was mostly enriched in factors involved in unfolded protein binding, regulating cuticle and extracellular matrix structure (Supplementary Fig. 3b), and many of these genes have been shown in a previous study to be targets of HSF-1 during HS[18]. Factors encoded by inactivated genes were

**Fig. 2 | Hormetic heat stress results in an ENDU-2-dependent reprogramming of the transcriptome after heat stress. a** Scatter Plot comparing the average gene expression changes upon 1 h HS (x-axis) and 4 h after HS (y axis) in WT animals. Each dot corresponds to a gene. Genes showing with a significant increase or decrease in transcript level (FC > 2, FDR < 0.01) only upon 1 h HS are shaded red or green respectively. Genes with a significantly increased or decreased transcript level only 4 h after HS are labeled in blue or yellow respectively. Genes activated or inactivated both during and after HS are labeled in purple or orange, respectively. Genes without a significant change (FC < 2 or FDR > 0.01) are shaded gray. **b** Scatter Plot comparing the average gene expression changes upon 1 h HS in WT (x-axis) and *endu-2(tm4977)* (y axis). Genes with a significantly increased or decreased transcript level (FC > 2, FDR < 0.01) upon 1 h HS in WT animals are in shaded red and green respectively. SCC: Spearman's correlation coefficient. **c** Scatter Plot comparing the average gene expression changes 4 h after HS in WT (x axis) and *endu-2(tm4977)* (y-

axis). Genes with a significant increased or decreased transcript level (FC > 2, FDR < 0.01) 4 h after HS in WT animals are in shaded dark blue and yellow respectively. **d** The Heat map showing the relative gene expression of three classes of post-HS responsive genes in WT and *endu-2(tm4977)* animals in HS and post-HS phase. The data used to create the heat map is summarized in Supplementary Data 2. On the right side of the heat map, a diagram illustrates the changes of gene expression that occurred within each class. **e** Top 5 enriched biological processes controlled by the post-HS responsive genes displaying a significant increase (upper panel) or decrease (lower panel). **f** RT-qPCR quantification of the relative transcript levels of representative HS and post-HS responsive genes in WT and *endu-2(tm4977)* animals at indicated conditions. Data are the mean ± SD of three biological replicates, each with three technical replicates, normalized to the mean expression levels of housekeeping gene *act-4*. *P* values were calculated using two-way ANOVA with Tukey's multiple comparisons test.

---

enriched in control of neuronal function (Supplementary Fig. 3c). 4 h after HS, WT animals showed 565 genes with increased and 109 with decreased transcript expression level (FC > 2, FDR < 0.01) (Fig. 2a, c and Supplementary Data. 1). We named these DEGs after HS "post-HS responsive genes". Intriguingly, only 18% (102/565) of the activated and 30% (33/109) of the inactivated genes in the post-HS period were also HS-responsive genes (Fig. 2a), suggesting that the alteration of the transcriptome in the post-HS phase is distinct from that during HS and the canonical HS response is quickly shut off after HS.

Based on their expression level without HS, directly upon 1 h HS and 4 h after HS, we grouped the "post-HS responsive genes" into three classes (Fig. 2d, Supplementary Data. 2). Class I genes showed increased (70/565) or decreased (28/109) expression in response to HS and displayed a slow reset after HS so that they were still significantly differently expressed compared to unstressed condition; Transcript level of the Class II genes were initially increased (32/565) or decreased (5/109) during HS and then further enhanced in post-HS phase; Class III contains the most post-HS responsive genes, which did not significantly respond to HS and only became active (463/565) or inactive (76/109) in the post-HS phase. We found that the Class III post-HS responsive genes did not overlap with those showing increased expression after prolonged HS[19] (Supplementary Fig. 3d), suggesting they probably only respond to recovery rather than an extended duration of stress exposure. The Additional GO term analysis suggests that functions of activated Class III genes were enriched in defense and immune response and IRE1-mediated unfolded protein response, while the inactivated transcripts were mostly enriched in regulating diverse metabolic processes (Fig. 2e). These data also implicate that, although HS response is generally rapidly reset to the normal level after HS, the altered expression level of some HS responsive genes can be maintained to certain extent (Class I genes) or further enhanced (Class II genes) after stress. In addition, a distinct transcriptional reprogramming in the post-HS phase arises (Class III genes).

Next, we asked whether *endu-2(tm4977)* animals have a similar HS or post-HS response as WT. We observed a high correlation of transcriptomic alteration between WT and *endu-2(tm4977)* animals during HS (Spearman correlation coefficient (SCC) = 0.75) (Fig. 2b), suggesting that *endu-2(tm4977)* and WT animals have similar HS response. This may explain why *endu-2(tm4977)* animals are similarly resistant to HS as WT. In contrast, there was a low correlation of fold change between WT and *endu-2(tm4977)* mutant (SCC = 0.38) in the post-HS phase (Fig. 2c) and expression alteration of 61% of genes in WT animals was *endu-2* dependent (Fig. 2d, Supplementary Data. 3). To further validate these data, we performed quantitative RT-PCR (qRT-PCR) to compare the relative abundance of selected HS responsive as well as post-HS responsive genes in WT and *endu-2(tm4977)* animals. The genes were chosen based on two primary criteria: First, they display the most significant changes in their transcript levels. Second, published primer sequences are available for qPCR quantification. The qRT-PCR results further confirmed that lack of *endu-2* did not impair activation of HS

responsive genes (*hsp-16.11* and *hsp-16.48*), as well as Class I genes (*hsp-16.2* and *hsp-70*) directly by HS (Fig. 2f). However, inactivation of *endu-2* resulted in significantly lower expression level of the selected genes of the all three classes of post-stress responsive genes 4 h after HS (Fig. 2f), further supporting the hypothesis that ENDU-2 is specifically involved in regulation of gene expression after HS.

Next, we asked whether the activated post-HS responsive genes indeed contribute to heat hormesis. A participation of the Class II gene *sqst-1*, which encodes the *C. elegans* homolog of SQSTM1 to activate autophagy, to heat hormesis has been reported previously[5]. The *hsp-16.2* gene encodes a chaperone that facilitates protein folding[20]. Both *zip-10* and *pqm-1* encode transcription factors that have been reported to function in immune response[21,22]. *irg-2* is considered to participate in innate immunity response based on its elevated expression level upon pathogen infection[23,24]. We performed RNAi to inactivate these selected Class I (*hsp-16.2*), Class II (*zip-10*) and Class III genes (*irg-2* and *pqm-1*) and tested the survival of animals upon a subsequent exposure to HS or CdCl2 after hormetic HS. RNAi knockdown of *hsp-16.2, zip-10* or *irg-2*, or *pqm-1* mutation abolished hormetic HS-induced resistance against HS and cadmium toxicity (Supplementary Fig. 4a-e), suggesting that the post-HS response genes play important roles in mediating beneficial effect of hormetic HS.

We additionally compared ENDU-2-dependent post-HS responsive genes with a published tissue-specific transcriptome study[25]. We found that intestinal specifically expressed genes were most dominant among the post-HS gene (Supplementary Fig. 5a and Supplementary Data 4). This correlates to our tissue specific rescue data with *endu-2* transgene (Fig. 1e), indicating that intestinal ENDU-2 may mediate beneficial effect of hormetic HS primarily in a cell-autonomous manner.

Our RNA-seq data additionally revealed that *endu-2(tm4977)* mutants had increased level of 1109 and decreased level of 64 transcripts compared to WT animals under normal growth conditions (Supplementary Data 1), implicating an overall negative impact of ENDU-2 on gene expression in the absence of HS. This group of normally repressed genes includes more than 40% (141 out of 346) of post-HS responsive genes that are activated by ENDU-2 after a HS (Supplementary Fig. 5b). qRT-PCR of selected genes validated the opposing functions of ENDU-2 under stress-free *vs.* post-HS conditions (Fig. 2f). We conclude that ENDU-2 is capable of influencing the transcript level of the same gene in both directions. How these two opposite functions are achieved remains unknown.

## ENDU-2 positively regulates gene expression at the transcriptional level after HS

Although ENDU-2 as a mRNA-binding endoribonuclease is known to decrease mRNA levels, our transcriptomic study indicates that ENDU-2 promotes expression of the post-HS responsive genes after HS. Therefore, we asked whether ENDU-2 acts at transcriptional or post-transcriptional level in the post-HS period. For this, we fused a 2.5 kb

genomic fragment upstream of *irg-2* start codon to mCherry to generate a transcriptional reporter strain *byIs296[irg-2p::mCherry]*, since *irg-2* is one of the most strongly up-regulated Class III genes by ENDU-2 after HS. Under reference growth conditions, mCherry expression was very low, and neither 1 h nor prolonged HS induced expression of mCherry (Fig. 3a and Supplementary Fig. 6a–c). However, 8 h after a 1 h HS, we observed a significantly increased mCherry expression, suggesting a transcriptional activation of *irg-2* specifically in the post-stress phase (Supplementary Fig. 6a-c). If ENDU-2 positively affects *irg-2* mRNA level at post-transcriptional level, inactivation of *endu-2* should not impair induction of this transcriptional reporter. Surprisingly, hormetic HS failed to activate *irg-2p::mCherry* reporter in *endu-2(tm4977)* mutant animals (Fig. 3a, c). In contrast, inactivation of *endu-2* did not affect HS-induced activation of the *hsp-16.2p::GFP* reporter (Fig. 3a, d). All these results collectively suggest that ENDU-2 specifically activates transcription of the post-HS responsive genes after HS.

Previous studies have shown the involvement of HSF-1, DAF-16, DAF-12 and SWSN-1 in heat hormesis-induced lifespan extension[6,26,27]. To investigate whether these factors affect HS or post-HS response, we used both *hsp-16.2* and *irg-2* transcriptional reporters to monitor their expression level upon RNAi knockdown of these transcriptional regulators. RNAi knockdown of *hsf-1, daf-16, or daf-12* strongly impaired transcriptional activation of *hsp-16.2* upon HS (Fig. 3b, e) but not of *irg-2* after HS (Fig. 3b, f). *swsn-1* RNAi, on the other hand, affected both *irg-2p::mCherry* and *hsp-16.2p::GFP* induction. In addition, lack of *swsn-1* abolished the beneficial effect of hormetic HS (Supplementary Fig. 7a, b), and strongly reduced *hsp-16.2* and *irg-2* mRNA levels both during and after HS (Supplementary Fig. 7c). *swsn-1* encodes a core component of the SWI/SNF chromatin remodeling complex[28], we further found that RNAi knockdown of the other core subunits of worm SWI/SNF complex *swsn-4* and *swsn-5* prevented activation of the *irg-2* transcriptional reporter (Supplementary Fig. 7d, e). As *C. elegans* neurons are refractory to RNAi, these observations additionally suggest that these factors mostly act out side of the neuronal system to control HS or post-HS response. These observations additionally indicate that heat hormesis relies on transcriptional reprogramming both during HS and after HS that are differently regulated and can be uncoupled from each other.

## ENDU-2 localizes in the nucleus and associates with chromatin

To gain a hint of how ENDU-2 could activate the post-HS response, we searched for ENDU-2-associated proteins in the post-HS phase with mass spectrometry (MS). For this, we immunoprecipitated ENDU-2::EGFP in our EGFP knock-in strain via CRISPR-Cas9 upon 1 h HS and 2 h recovery and identified co-immunoprecipitated proteins with quantitative MS ($\log_2$LFQ intensity >1). In line with our previous work revealing ENDU-2 as a secreted endoribonuclease, our MS result showed an association of ENDU-2 with multiple factors involved in ER-Golgi transport and proteins in the extracellular matrix (Supplementary Data 5 and Supplementary Fig. 8). An unexpected finding was that multiple nuclear proteins, as well as factors controlling nuclear import/export, were also identified as ENDU-2 interactors, raising the question whether a portion of ENDU-2 may be localized in the nucleus. To test whether ENDU-2 exists in the nucleus, we performed a cellular fractionation and could detect ENDU-2::EGFP both in the nucleus and cytoplasm (Fig. 4a). Next, we used more sensitive immunofluorescence assay to visualize nuclear localization of ENDU-2::EGFP protein. The antibody staining procedure would also abolish the gut autofluorescence to help detecting weakly expressed ENDU-2::EGFP protein in the intestine. Confocal microscopy showed that ENDU-2::EGFP in our EGFP CRISPR knock-in strain was localized both in the cytosol and in the nucleus under unstressed conditions (Fig. 4b). In addition, ENDU-2::EGFP formed condensate-like granule structures and some of them colocalized with DAPI-stained DNA. As the antibody stain against histone H3 completely overlapped with the DAPI signal

(Supplementary Fig. 9), ENDU-2::EGFP may associate with chromatin. In line with this, ENDU-2::EGFP was co-immunoprecipitated with histone 3 (H3) in an RNA-independent manner, as RNA digest with RNase A did not interfere with the protein interaction (Fig. 4c). Next, we tested whether HS affects the subcellular localization of ENDU-2. We found that 1 h HS at 35 °C resulted in a slightly reduced ENDU-2::EGFP protein level in the cytoplasm but not in the nucleus (Fig. 4f, g). However, colocalization of ENDU-2::EGFP with DAPI-stained DNA was enhanced and this could be maintained 2 h after HS (Fig. 4d, e). This observation indicates that ENDU-2 displays a dynamic localization on chromatin that might be linked to its role in transcriptional regulation.

As HS alters the distribution of ENDU-2 in the nucleus, we tested whether HSF-1 or the SWI/SNF complex could be involved in the transmission of the stress signal to ENDU-2. We found that RNAi knockdown of neither *hsf-1* nor *swsn-1* interfered with chromatin localization of ENDU-2 during and after HS (Supplementary Fig. 10a–c), suggesting that an unknown mechanism might signal ENDU-2 to change its nuclear distribution. Next, we tested whether ENDU-2 affects SWSN-1 localization and found that SWSN-1::GFP exhibited constant nuclear localization regardless of the presence of heat stress or the *endu-2* gene (Supplementary Fig. 7f). These results suggest that nuclear localization of ENDU-2 and SWSN-1 is probably independent of each other, although monitoring their subcellular localization does not provide enough information whether one of these two proteins affects the association of the other with respective chromatin loci.

## Contribution of RNA-binding and -cleavage activities of ENDU-2 to heat hormesis

ENDU-2 binds to mRNA and is capable of cleaving RNA. An E to Q substitution of amino acid at the position 454 (E454Q) abolishes its RNA-cleavage activity without interfering with RNA-binding, whereas an E to Q substitution at position 460 (E460Q) impairs RNA-binding and therefore also cleavage capacity of ENDU-2[15]. We asked whether these two ENDU-2 mutant variants might behave different from wild-type ENDU-2 regarding their contributions to heat hormesis. Transgenically expressed wild-type ENDU-2::EGFP but not ENDU-2(E454Q)::EGFP or ENDU-2(E460Q)::EGFP, was able to improve survival of *endu-2(lf)* animals upon a subsequent HS and Cd$^{2+}$ treatment after exposure to hormetic HS (Fig. 5a–c). In addition, neither ENDU-2(E454Q), nor ENDU-2(E460Q) behaved similar to wild-type ENDU-2 to maintain *hsp-16.2* mRNA levels and activate *irg-2* after HS in *endu-2(lf)* animals (Fig. 5d, e). Next, we examined whether ENDU-2(E454Q)::EGFP and ENDU-2(E460Q)::EGFP are chromatin associated. Cell fractionation experiments revealed that both ENDU-2(E454Q)::EGFP and ENDU-2(E460Q)::EGFP were present in the nucleus (Supplementary Fig. 11a) and they both could be co-immunoprecipitated with histone H3 (Supplementary Fig. 11b). However, ENDU-2(E454Q)::EGFP in the intestine did not show obvious condensate-like granule structures that colocalized with DAPI-stained DNA either under normal growth condition or upon HS (Supplementary Fig. 11c, d). In contrast, ENDU-2(E460Q) formed condensates colocalizing with DNA and this could be further enhanced by HS (Supplementary Fig. 11c, e). Although we do not know why ENDU-2(E454Q) is incapable of forming chromatin-associating condensates in the intestinal cells, the observation with ENDU-2(E460Q) nevertheless indicates that the dynamically changed chromatin association of ENDU-2 upon HS does not rely on its RNA-binding activity, similar to the RNA-independent interaction with H3 (Fig. 4c).

## ENDU-2 activates RNA polymerase II to trigger post-HS response

Our MS interactor study revealed the interaction of ENDU-2 with multiple chromatin components (Supplementary Data 5 and Supplementary Fig. 8). ENDU-2 interacted with three proteins of the SWI/SNF complex: SWSN-1, SWSN-4, and ISW-1, and 3 subunits of RNA

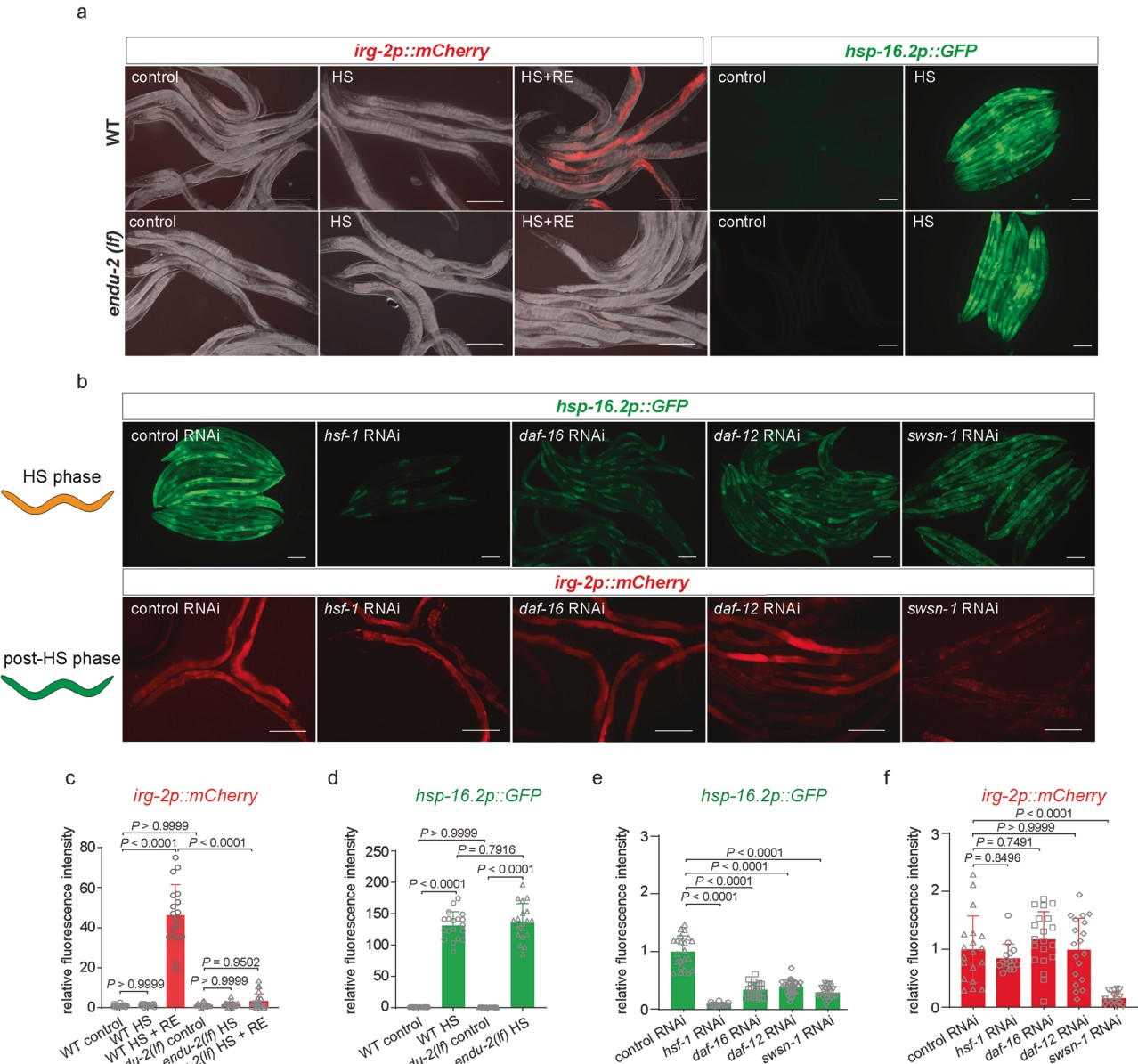

**Fig. 3 | ENDU-2 regulates gene expression at the transcriptional level after HS.**
**a** Fluorescence micrographs of transgenic *byIs296[irg-2p::mCherry]*, *endu-2(tm4977);byIs296[irg-2p::mCherry]*, *ncIs17[hsp-16.2p::GFP]* and *endu-2(tm4977);ncIs17[hsp-16.2p::GFP]* animals. The mCherry intensity was detected in day 1 adult animals subjected to 1 h HS or 1 h HS followed by 16 h recovery at 20 °C. The GFP intensity was detected in day 1 adult animals subjected to 4 h HS without recovery. RE: recovery. Scale bar = 200 μm. **b** Fluorescence micrographs of transgenic *ncIs17[hsp-16.2p::GFP]* and *byIs296[irg-2p::mCherry]* animals fed with control HT115 bacteria carrying L4440 empty vector (control) or those expressing dsRNA targeting *hsf-1*, *daf-16*, *daf-12* or *swsn-1*. Upper panel: GFP was detected in day 1 adult animals subjected to 4 h HS without recovery. Lower panel: mCherry was detected in animals subjected to 1 h HS at day 1 adulthood followed by 16 h recovery at 20 °C. Scale bar = 200 μm. **c** Quantification of relative mCherry fluorescence intensity in *byIs296[irg-2p::mCherry]* (WT background) and *endu-2(tm4977);byIs296[irg-2p::mCherry]* (*endu-2* mutant background) animals under different conditions. WT background control (20 °C): *n* = 19. WT background with 1 h HS: *n* = 18. WT background after HS (HS + RE): *n* = 18. *endu-2* mutant background control: *n* = 15. *endu-2* mutant background with 1 h HS: *n* = 18. *endu-2* mutant background after HS (HS + RE): *n* = 23. Data are the mean ± SD, each dot is the

relative fluorescence intensity per animal. *P* values were calculated using one-way ANOVA with Tukey's multiple comparisons test. **d** Quantification of relative GFP fluorescence intensity in *gpIs1 [hsp-16.2p::GFP]* (WT background) and *endu-2(tm4977);gpIs1 [hsp-16.2p::GFP]* (*endu-2* mutant background) animals under different conditions. WT background control: *n* = 18. WT background with 4 h HS: *n* = 19. *endu-2* mutant background control: *n* = 18. *endu-2* mutant background with 4 h HS: *n* = 19. Data are the mean ± SD, each dot represents the relative fluorescence intensity per animal. *P* values were calculated using one-way ANOVA with Tukey's multiple comparisons test. **e** Quantification of relative GFP fluorescence intensity links to the upper panel of **b**. RNAi control: *n* = 22. *hsf-1* RNAi knock down: *n* = 17. *daf-16* RNAi knock down: *n* = 25. *daf-12* RNAi knock down: *n* = 33. *swsn-1* RNAi knock down: *n* = 23. Data are the mean ± SD, each dot is the relative fluorescence intensity per animal. *P* values were calculated using one-way ANOVA with Tukey's multiple comparisons test. **f** Quantification of relative mCherry fluorescence intensity links to the lower panel of **b**. RNAi control: *n* = 19. *hsf-1* RNAi knock down: *n* = 15. *daf-16* RNAi knock down: *n* = 20. *daf-12* RNAi knock down: *n* = 19. *swsn-1* RNAi knock down: *n* = 16. Data are the mean ± SD, each dot is the relative fluorescence intensity per animal. *P* values were calculated using one-way ANOVA with Tukey's multiple comparisons test.

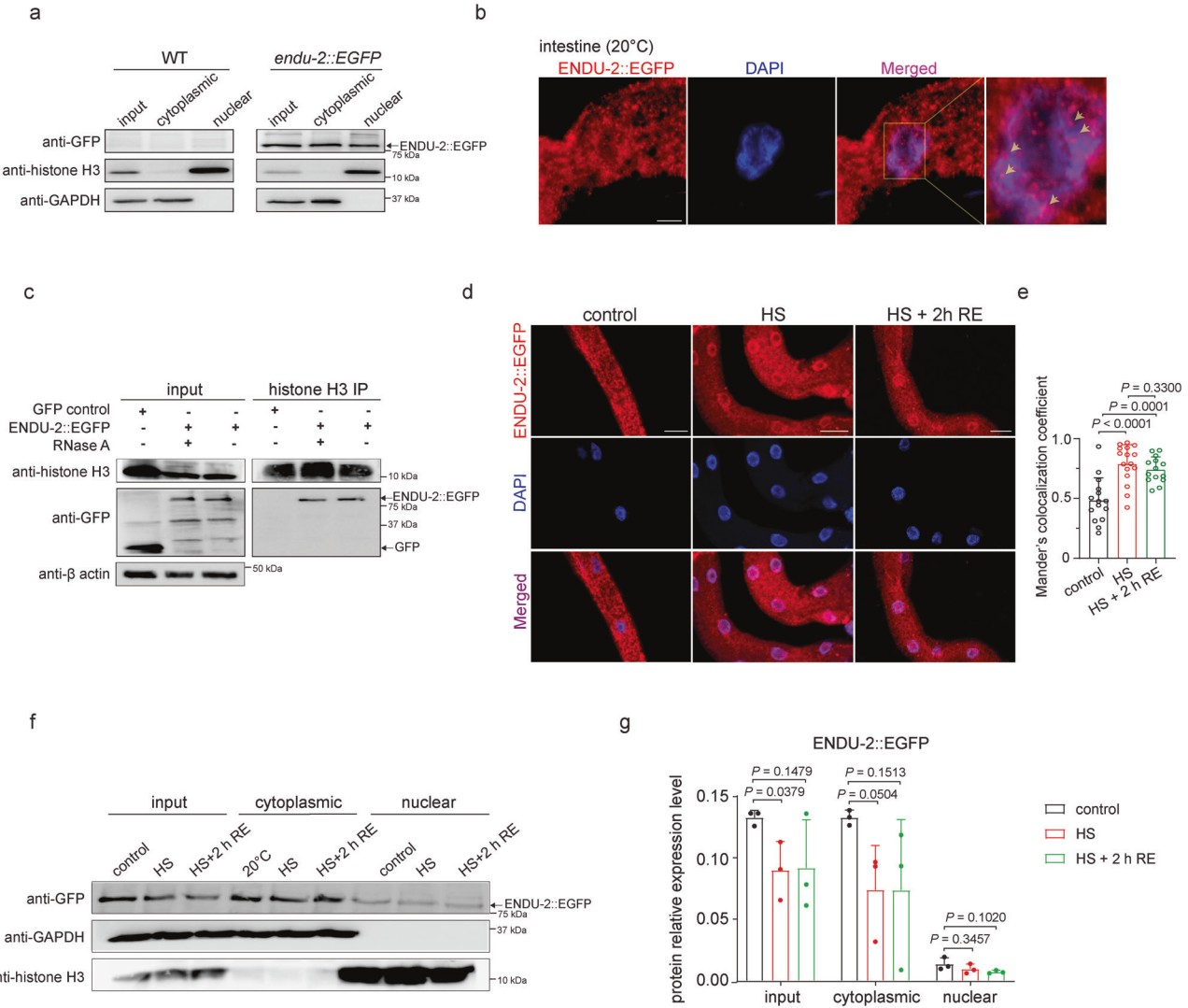

**Fig. 4 | ENDU-2 localizes in the nucleus and associates with chromatin.**
**a** Western blot detection of ENDU-2::EGFP in different cellular fractions. WT N2 serves as the negative control for ENDU-2::EGFP detection. ENDU-2::EGFP can be detected in whole cell lysis (input), cytoplasmic fraction, and nuclear fraction. Histone H3 and GAPDH served as positive controls for nuclear and cytoplasmic fractions respectively. $N = 3$ independent experiments. **b** Immunofluorescence staining of intestinal ENDU-2::EGFP with GFP antibody in *endu-2(by190[endu-2::EGFP])* CRISPR knock-in animals at 20 °C. Red: ENDU-2::EGFP, blue: DAPI-stained DNA. The white arrows point to overlaps between ENDU-2::EGFP and DAPI-stained DNA in pink. $n = 25$ animals, $N = 4$ independent experiments. Scale bar = 5 μm.
**c** ENDU-2::EGFP is co-immunoprecipitated with histone H3 independent of RNA. A transgenic *Is[sod-3p::gfp]* strain serves as a negative control to exclude the interaction between GFP and histone H3. $N = 3$ independent experiments.
**d** Immunofluorescence staining of ENDU-2::EGFP with GFP antibody in *endu-2 (by190[endu-2::EGFP])* CRISPR knock-in animals subjected to 1 h HS or 1 h HS

followed by 2 h recovery. Scale bar = 20 μm. **e** Quantification of overlaps between the fluorescence of ENDU-2::EGFP and DAPI-stained DNA with Mander's colocalization method. $N = 3$ independent experiments. Animals at control (20 °C): $n = 15$. Animals upon HS: $n = 16$. Animals after HS (HS + RE): $n = 14$. Data are the mean ± SD, each dot represents the Mander's colocalization coefficient of one intestinal nucleus. $P$ values were calculated using two-tailed unpaired Student $t$ tests.
**f** Western blot detection of ENDU-2::EGFP in different subcellular fractions of *endu-2(by190[endu-2::EGFP])* animals that were unstressed, HS stressed (35 °C, 1 h) and HS stressed followed by recovery (20 °C, 2 h). Histone H3 and GAPDH served as internal controls for nuclear and cytoplasmic fractions respectively. $N = 3$ independent experiments. **g** Quantification of relative protein level of ENDU-2::EGFP in different cellular fractions under different conditions via Western blot detection in **f**. Relative protein level was normalized to internal controls. Data are the mean ± SD, and $P$ values were calculated using one-way ANOVA with Tukey's multiple comparisons test.

polymerase II (Pol II) (RPB-2, RPB-5 and RPB-7). We confirmed physical interactions between ENDU-2 and RPB-2 as well as AMA-1, the Pol II subunit A via co-IP (Fig. 6a–c). Interaction between Pol II and ENDU-2::EGFP seemed to be largely RNA dependent as RNase A treatment strongly reduced the amount of co-immunoprecipitated proteins (Fig. 6a–c and Supplementary Fig. 12). In addition, ENDU-2(454Q)::EGFP and ENDU-2(E460Q):EGFP failed to co-immunoprecipitate with AMA-1 (Fig. 6d). Combined with the fact that ENDU-2(E460Q)::EGFP displayed a chromatin localization similar to ENDU-2(WT)::EGFP, these observations suggest that RNA-independent chromatin association of ENDU-2 is insufficient for its

interaction with Pol II and ENDU-2 probably needs participation of certain RNA species to bind to Pol II. Next, we addressed whether ENDU-2 might regulate Pol II function during or after HS. For this, we performed Pol II ChIP-qPCR and tested Pol II binding on promoters and gene bodies of the most significantly up-regulated post-stress responsive genes: *hsp-70* (Class I), *C25F9.2* (Class II), *C25F9.12* (Class III) and *irg-2* (Class III). We considered that transcription of these genes by Pol II might occur earlier than 4 h after HS, the time point to study our post-stress transcriptome. Therefore, we performed Pol II ChIP-qPCR with animals that were non-stressed, 1 h stressed without recovery, stressed and recovered for 1 h, 2 h, and 4 h. 1 h HS exposure provoked

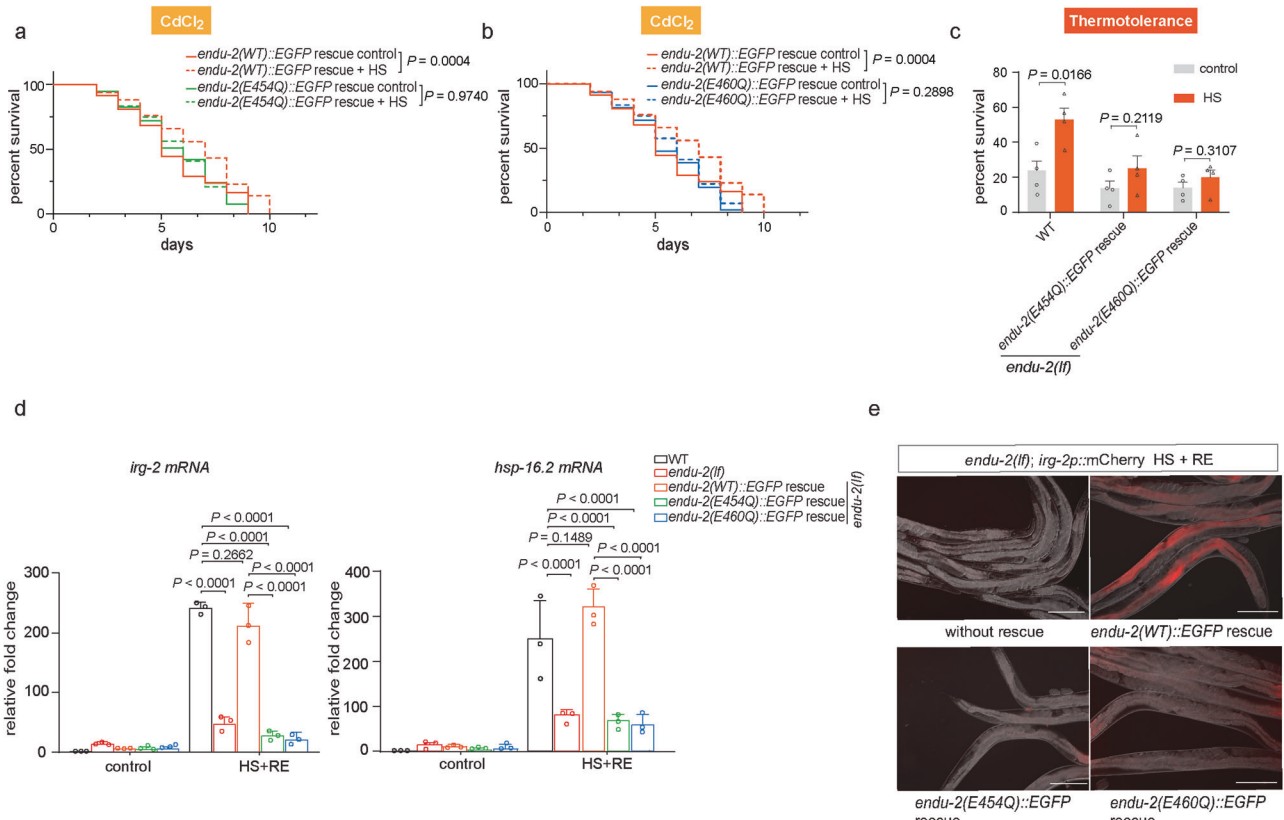

**Fig. 5 | Contribution of RNA-binding and -cleavage activities of ENDU-2 to heat hormesis. a**, **b** Survival curves of *endu-2(tm4977);byEx1375[endu-2p::endu-2::EGFP]* (*endu-2(WT)* rescue), *endu-2(tm4977);byEx1492[endu-2p::endu-2(E454Q)::EGFP::3x-FLAG]* (*endu-2(E454Q)* rescue) and *endu-2(tm4977);byEx1495[endu-2p::endu-2(E460Q)::EGFP::3xFLAG]* (*endu-2(E460Q)* rescue) animals exposed to CdCl₂ treatment. Animals were subjected to 1 h HS followed by 4 h recovery at 20 °C before they were exposed to 0.25 mM CdCl₂. Mean survival for each group: *endu-2(WT)* rescue without HS: 5.5 days, *n* = 104. *endu-2(WT)* rescue with HS: 6.6 days, *n* = 100. *endu-2(E454Q)* rescue without HS: 5.7 days, *n* = 110. *endu-2(E454Q)* rescue with HS: 5.8 days, *n* = 96. *endu-2(E460Q)* rescue without HS: 5.5 days, *n* = 103. *endu-2(E460Q)* rescue with HS: 5.8 days, *n* = 85. Shown are pooled data of *N* = 3 independent experiments. *P* values were calculated using Log-rank (Mantel−Cox) test. **c** Thermotolerance of WT, *endu-2(E454Q)* rescue and *endu-2(E460Q)* rescue animals after HS at 35 °C for 8 h. For thermotolerance after hormetic HS, day one adult animals were incubated at 35 °C for 1 h, followed by a 12 h recovery at 20 °C before exposure to the subsequent HS for 8 h. Survival rates for each group: WT control: (22.5 ± 7.2) %, *n* = 112; WT HS: (52.9 ± 8.3) %, *n* = 121; *endu-2(E454Q)* rescue control: (13.5 ± 6.4) %, *n* = 118; *endu-2(E454Q)* rescue HS: (25.0 ± 8.2) %, *n* = 121; *endu-2(E460Q)* rescue control: (13.9 ± 6.2) %, *n* = 106; *endu-2(E460Q)* rescue HS: (19.9 ± 6.3) %, *n* = 102, *N* = 4. Data are the mean ± SEM *P* values were calculated with two-tailed multiple unpaired *t* test. **d** RT-qPCR quantification of the relative mRNA levels of *irg-2* and *hsp-16.2* in WT, *endu-2(tm4977)*, *endu-2(WT)* rescue, *endu-2(E454Q)* rescue and *endu-2(E460Q)* rescue animals under the indicated conditions. Data are the mean ± SD of three biological replicates, each with three technical replicates, normalized to the mean expression levels of house-keeping gene *act-4*. *P* values were calculated using two-way ANOVA with Tukey's multiple comparisons test. **e** Fluorescence micrographs of transgenic *endu-2(tm4977);byIs296[irg-2p::mCherry]* animals with *endu-2(WT)* rescue, *endu-2(E454Q)* rescue and *endu-2(E460Q)* rescue. mCherry intensity was measured in animals subjected to 1 h HS in day 1 adulthood followed by 16 h recovery at 20 °C. *N* = 3 independent experiments. *n* > 20 animals for each strain. Scale bar = 200 μm.

Pol II binding to promoter and gene body of *hsp-70* (Class I) (Fig. 6e) and this increased binding was already lost within 2 h after HS. For the Class II gene *C25F9.2*, Pol II showed increased binding to the promoter and gene body both during HS and 2 h after HS. In contrast, increased Pol II binding to the promoters and gene bodies of the Class III genes *irg-2* and *C25F9.12* only occurred 1 h after HS but not during HS. 2 h after HS, we detected even more Pol II association with these two Class III genes but this completely vanished another 2 h later. Notably, the lack of ENDU-2 significantly impaired Pol II binding on the promoters and gene bodies of the Class II gene both during and after HS. For the Class III genes, ENDU-2 only acted in the post-stress phase. These results together implicate an essential role of ENDU-2 to activate Pol II to induce post-HS response.

### HS activates binding of ENDU-2 to the promoter regions of the Class II and Class III post-HS responsive genes

Our results so far suggest a model in which ENDU-2 might interact with Pol II to facilitate transcription of the post-HS responsive genes. If this hypothesis is true, ENDU-2 should also bind to the same chromatin region. To test this, we performed ENDU-2::EGFP ChIP and quantified co-immunoprecipitated respective regions under different conditions via qPCR. We found that HS already stimulated ENDU-2::EGFP to bind to the promoter regions of each selected Class II (*C25F9.2*) and Class III (*irg-2* and *C25F9.12*) post-HS responsive genes and this increased binding was maintained 2 h after HS (Fig. 6f). In contrast, ENDU-2 did not increase its binding to the promoter of the Class I gene *hsp-70* either during or after HS. Moreover, we did not detect elevated ENDU-2 association with the gene bodies of these genes, arguing against an unselective binding of ENDU-2 to chromatin. Taken together, these results suggest that binding of ENDU-2 to the promoters of the post-HS responsive genes takes place prior to recruitment of Pol II for their transcriptional initiation. In addition, ENDU-2 might affect the Class I genes at the post-transcriptional level.

## Discussion

Hormesis refers to a unique response characterized by a biphasic reaction to increased quantities of substances or stimuli that are typically harmful at high doses but exhibit beneficial effects within a

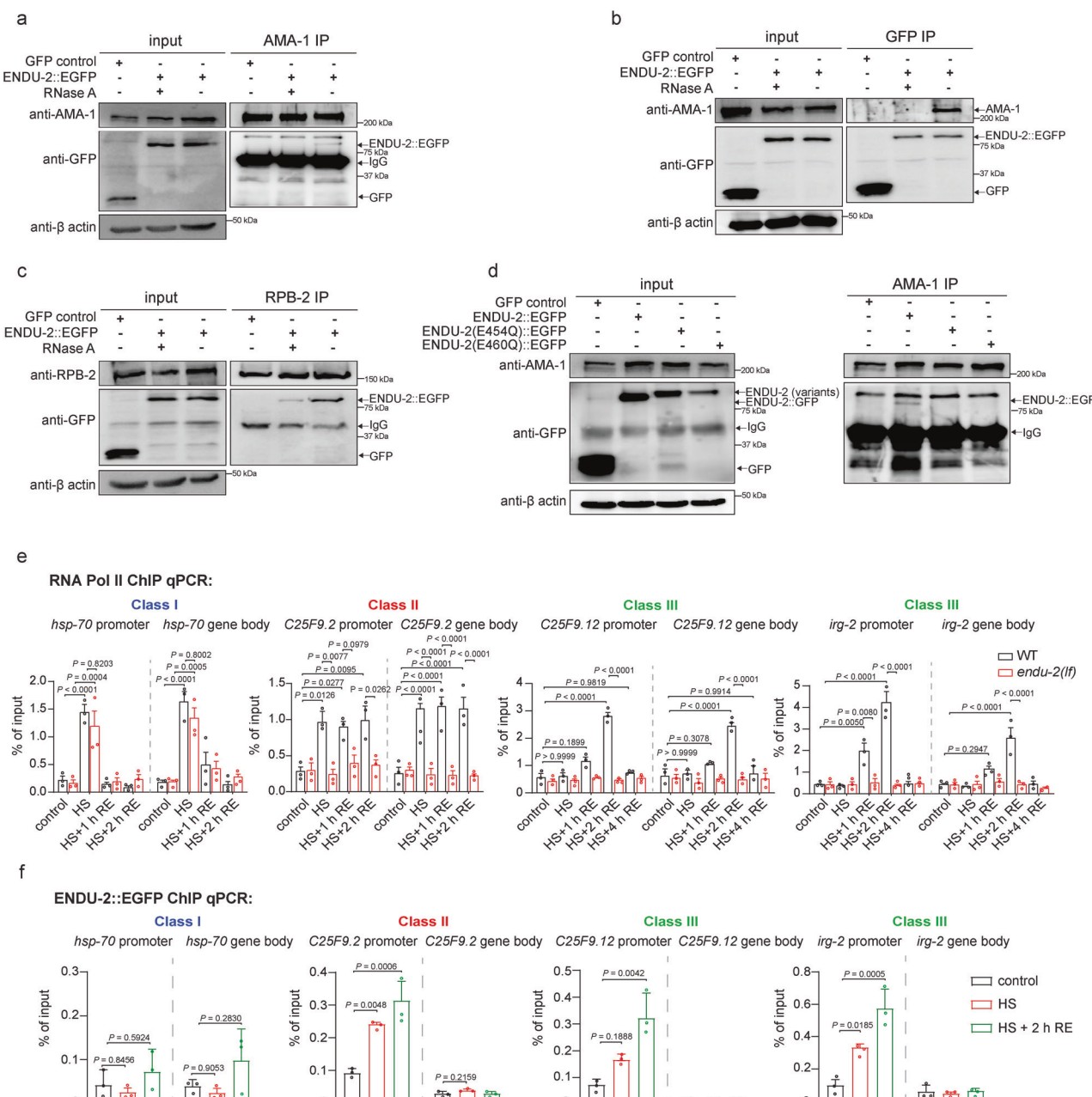

**Fig. 6 | ENDU-2 interacts with RNA polymerase II to activate Pol II after HS.**
**a**, **b** Co-immunoprecipitation of ENDU-2::EGFP with AMA-1 (Pol II subunit A) is RNA-dependent. Interaction between ENDU-2::EGFP and AMA-1 can be detected with either AMA-1 IP (**a**) or GFP IP (**b**). The interaction was detected in *endu-2(by190[endu-2::EGFP])* CRISPR knock-in day 1 adult animals. Transgenic *Is[sod-3p::GFP])* strain serves as a negative control to check potential interaction between GFP and AMA-1. *N* = 3 independent experiments. **c** Co-immunoprecipitation of ENDU-2::EGFP with RPB-2 (Pol II subunit B) is largely RNA dependent. The interaction was detected in day 1 adults of *endu-2(by190[endu-2::EGFP])* CRISPR knock-in animals. *N* = 3 independent experiments. **d** ENDU-2::EGFP but not ENDU-2(E454Q)::EGFP or ENDU-2(E460Q)::EGFP is co-immunoprecipitated with AMA-1. The interaction was

detected in *endu-2(by190[endu-2::EGFP])*, transgenic *endu-2(tm4977);byIs241[endu-2p::endu-2(E454Q)::EGFP]::3xFLAG]* and *endu-2(tm4977);byIs240[endu-2p::endu-2(E460Q)::EGFP]::3xFLAG]* day 1 adult animals. *N* = 3 independent experiments. **e** Pol II (AMA-1) ChIP-qPCR to compare Pol II binding on the promoters and gene bodies of selected post-HS responsive genes in WT and *endu-2(tm4977)* animals under respective conditions. *N* = 3 independent experiments. Data are the mean ± SD, *P* values were calculated using two-way ANOVA with Tukey's multiple comparisons test. **f** ChIP-qPCR analysis of ENDU-2::EGFP binding on the promoters and gene bodies of selected post-HS responsive genes in *endu-2(by190[endu-2::EGFP])* CRISPR knock-in animals. *N* = 3 independent experiments. Data are the mean ± SD, *P* values were calculated using one-way ANOVA with Tukey's multiple comparisons test.

specific low-dose range known as the hormetic zone. The prevailing theory that supports the advantageous outcomes is the over-compensation of defensive responses. Consequently, extensive research in hormesis has primarily concentrated on elucidating how low doses of toxins or stressors can activate various stress responses when exposed to stress. One classical illustration of hormesis is the induction of gene expression of heat-shock proteins and autophagy

regulators by hormetic HS[2,5,6], resulting in lifespan extension and improved stress resistance in *C. elegans*. Here we investigate the long-lasting influence of a hormetic HS in *C. elegans* and discover a transcriptional reprogramming in the post-stress phase that contributes to durable physiological alteration of animals (Fig. 7). We show that, while the acute HS response is composed of a rapid transcriptional activation and inactivation of a similar number of genes the subsequent

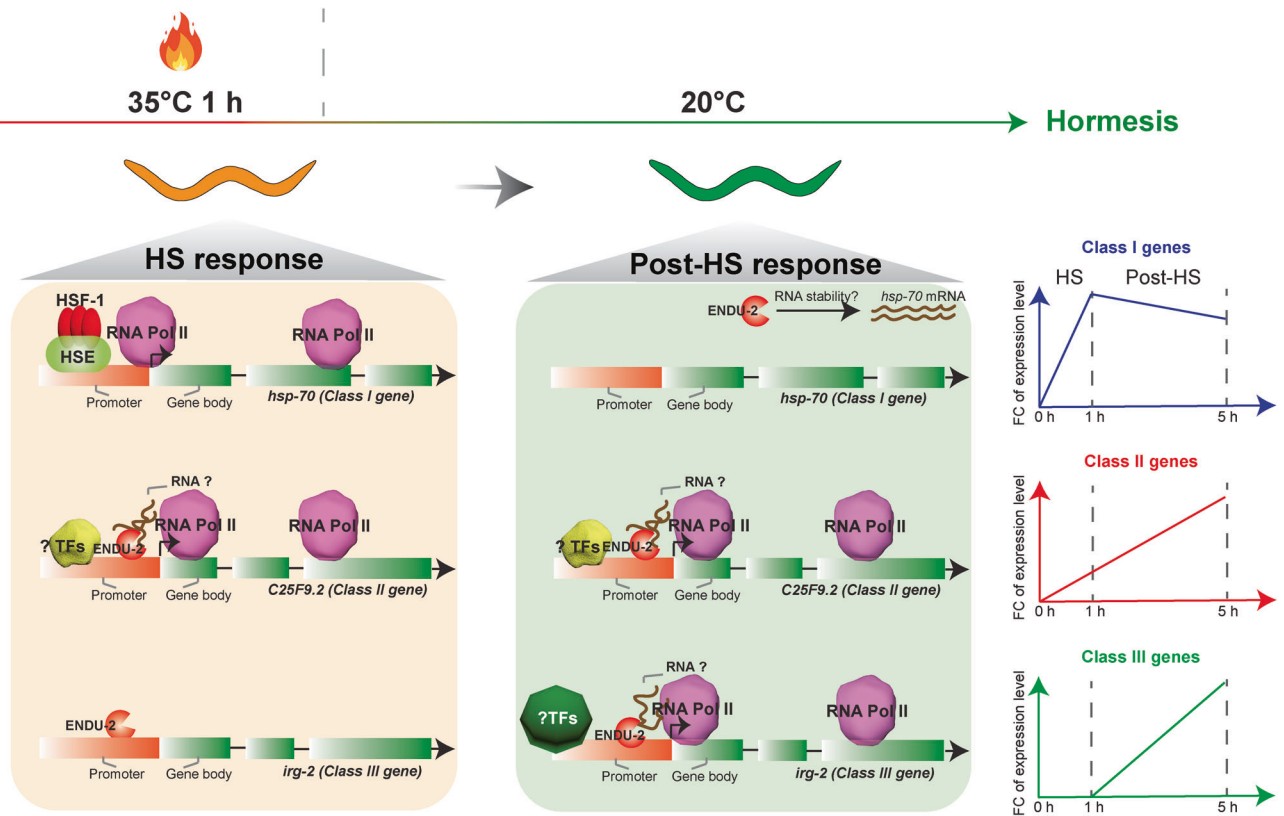

**Fig. 7 | Working model for transcriptional alternations of post-HS responsive genes.** Effective heat hormesis requires both HS- and post-HS responses, two distinct transcriptional reprogramming during and after HS. According to their temporal expression dynamics, the post-HS responsive genes are divided into three classes (right): Class I genes display altered transcript levels during HS that are reset slowly after HS, while reset of the most other HS responsive genes occurs rapidly after HS; Class II genes show altered expression levels during HS and these are further enhanced in the post-HS phase; Class III genes do not significantly respond to HS and only alter their expression level after HS. During HS, ENDU-2 increases its binding to the promoters of Class II and III post-HS responsive genes and promotes the recruitment of Pol II to the Class II genes. After HS, ENDU-2 maintains its association with the promoters of its target genes and activates Pol II also for the Class III genes. How ENDU-2 maintains the expression level of Class I targets is currently unknown.

post-HS response is regulated differentially and primarily results in the activation of genes, many of them linked to innate immunity. In addition, the differential control of HS and post-HS responses allows them to be uncoupled from each other: while HSF-1, DAF-12 and DAF-16 mainly affect the HS response during stress exposure, ENDU-2 acts later. Our results also indicate that the HS response alone determines survival of animals upon continuous HS, but long-term influence of a transient HS, such as heat hormesis relies on cooperation the both transcriptional reprogramming during and after HS[5,6]. It is important to emphasize that our study has only surveyed transcriptomic alteration at one time point after HS and our Pol II ChIP results strongly implicate a highly dynamic nature of the post-HS response. Therefore, we anticipate that our finding is only a glimpse at the entire picture of the post-HS response that is probably composed of chain reactions and feed-back loops of transcription factors, as well as epigenetic regulators at different time points after stress exposure. Detailing the spatiotemporal integration of these regulatory mechanisms will be fundamental to understand long-standing impacts of stress on cellular functions. While specific biological roles of most other ENDOU family members are not well known, all available studies in *C. elegans* suggest ENDU-2 to function in stress responses[11–14]. The N-terminal signal peptides for ER targeting makes ENDU-2 to promising candidates to coordinate systemic stress responses in a multicellular organism[15]. Our tissue specific *endu-2* rescue experiment and tissue specificity analysis of the post-HS responsive genes suggests intestine as the primary acting site of ENDU-2 and its targeting genes in mediating heat hormesis, auguring for a cell-autonomous activity of ENDU-2. In addition, ENDU-2 protein is detected in the nucleus of the intestinal

cells. Combined with the other indications for its cell-autonomous roles[12,13,15], one would ask how could a protein designated for ER-Golgi secretory pathway end up in cytoplasm and nucleus? Despite of lacking any experimental evidence yet, we propose that ENDU-2 might be exported out of the ER into the cytosol. One candidate of such a protein export machinery is the endoplasmic-reticulum-associated protein degradation (ERAD) pathway which targets misfolded proteins of the ER for ubiquitination and subsequent proteasome-mediated degradation in the cytoplasm. Several components of ERAD were indeed identified as ENDU-2 interactors according to our proteomics study, including CDC-48.2 (*Ce*. P97), SEL-1, and ENPL-1 (*Ce*. HSP90B1). Although we are not aware of any example for exporting functional proteins by ERAD, some viruses do hijack ERAD to penetrate the ER membrane to reach the cytoplasm[29]. Another possibility could be insufficient ER targeting of ENDU-2 by the signal peptide recognition particle (SRP) or impaired translocation of ENDU-2 peptide across the ER membrane, eventually resulting in cytoplasmic retention of the ENDU-2 protein. A mechanistic understanding distribution of ENDU-2 in- and outside of the ER-Golgi secretory pathway will elucidate how ENDU-2 coordinates its cell-autonomous and -non-autonomous activities to adjust stress response at the organismal level.

ENDU-2 and the other ENDOU family members have so far predominantly been linked to controlling gene expression via affecting RNA processing or stability. Here, we discover an unexpected role of ENDU-2 in the nucleus that is correlated with its association with chromatin, RNA and Pol II. Although our protein interaction and ChIP-qPCR experiments cannot distinguish between direct and indirect binding of ENDU-2 to Pol II or DNA, an association of ENDU-2 with

histone H3 was RNA-independent, whereas Pol II binding predominantly required the presence of RNA. This Pol II association in principle could indicate a co-transcriptional interaction with nascent mRNA and Pol II during transcription. However, based on our data, we rather favor ENDU-2 to be a regulator of transcription initiation, since ENDU-2 not only activated a reporter gene containing only the promoter of its downstream target gene *irg-2*, but also promoted recruitment of Pol II at the promoter of *irg-2* after HS. Notably, binding of ENDU-2 to the promoters of Class III post-HS responsive genes occurred already during HS, prior to recruitment of Pol II. It is currently unknown how such consecutive binding of ENDU-2 and Pol II may be controlled. This might include yet undefined RNA species that mediate the association between ENDU-2 and Pol II, or the contribution of an inhibitor preventing their interaction during HS. Although ENDU-2 binds to promoter regions of its target genes, our proteomics study did not identify any transcription factors as ENDU-2 interactors. Instead, factors affecting chromatin architecture, including three components of SWI/SNF chromatin remodeling complex (SWSN-1, SWSN-4, and ISW-1), were found to be associated with ENDU-2. As inhibition of the SWI/SNF complex inactivated both HS and post-HS responses, ENDU-2 probably acts together with it to improve accessibility of Pol II to the genomic loci of the post-HS responsive genes.

A recent large-scale RBP ChIP-seq analysis revealed widespread presence of RBPs in active chromatin regions, especially promoters of actively transcribed genes, in the human genome[30]. We propose that the active participation of ENDU-2 in transcriptional control might be pertinent to the concept of liquid-liquid phase separation. Despite the apparent absence of any low-complexity structural domain to facilitate phase separation, wild-type ENDU-2, but not RNA-cleavage deficient ENDU-2(E454Q) formed condensate-like structures both in the cytosol and in the nucleus. Whether these condensate-like structures are derived from phase separation and whether they participate in the post-stress response will further determine our understanding of the mechanistic details of ENDU-2-mediated transcriptional reprogramming in the post-stress phase. In such a scenario, interactions among ENDU-2, RNA, putative additional transcriptional regulators, and DNA segments are expected to form specific regulatory subareas to affect the DNA-binding ability of Pol II.

## Methods

### Strains
Strains were maintained at 15 °C and using *Escherichia coli* (*E. coli*) OP50 as a food source[31]. Animals used for experiment were cultured at 20 °C from embryonic stage. For all RNAi experiments, RNAi was initiated from L4 stage via feeding with *E. coli* HT115 stains carrying L4440 plasmids expressing respective double-stranded RNA against the target genes. F1 progeny was used for the respective analyses. The complete list of *C. elegans* strains used in this study is provided in Supplementary Data 6. The *C. elegans* N2 (Bristol) strain was used as wild-type in all experiments in this study. All *endu-2(tm4977)* animals were the early generations (G2-G4) from BR7295 *endu-2(tm4977);byEx1375[endu-2p::endu-2::EGFP;myo-2p::mCherry]* whose ancestor (G1) had lost the transgene. Sterile phenotype will appear in some *endu-2(tm4977)* mutant after loss of *endu-2::EGFP* transgene for about 10 generations at 20 °C. The *endu-2(tm4977)* mutant has a deletion that covers the part of the *endu-2* promoter, the whole 5'UTR to part of the third exon. Another allele *endu-2(by188)* used in this study contains a 20 bp deletion in the first exon that results in an early stop codon in the exon 1. Therefore, both *tm4977* and *by188* are considered to be strong loss of function or null mutants.

### Plasmids
To construct an mCherry transcriptional fusion reporter of *irg-2* (pBY4201), a 2723 bp genomic fragment upstream of the *irg-2* ATG was amplified with PCR and inserted into pmCherry-N1 vector with *Eco47III/*

*BglII* sites. Constructs expressing *endu-2p::endu-2::EGFP* (pBY3800), *endu-2p::endu-2(E454Q)::EGFP::3xFlag* (pBY3897) and *endu-2p::endu-2(E460Q)::EGFP::3xFlag* (pBY3898) are described in ref. 15.

### Lifespan analysis
Lifespan assay was carried out on 6 cm NGM plates seeded with *E. coli* OP50 bacteria and ~30–40 animals per plates. As *endu-2(tm4977)* animals display a strong egg-laying defect due to abnormal vulval development, agar plates containing 100 μM 5-fluorodeoxyuridine (Sigma-Aldrich) were used during the first seven days of adulthood to avoid internal hatching. For hormetic heat stress, L4 animals were selected and incubated at 35 °C for 1 h 18 h later (day 1 adult). For an experiment with a second hormetic heat stress, animals were additionally incubated at 35 °C for 1 h on day 3 of adulthood. Except those shown in Fig. 1b and Supplementary Fig. 1a, animals for lifespan with hormetic HS were exposed to 35 °C for 1 h on day 1 and day 3 adulthood stage. Animals were identified as dead if they failed to respond to prodding with a platinum wire. We used incubation at 36 °C for 45 min as heat hormesis for lifespan without FUDR and animals died due to internal hatching of larvae were censored from the assay.

### Thermotolerance
Thermotolerance assay was performed as described in ref. 2,32. 30–40 mid-L4 animals raised on OP50 at 20 °C were selected. After 18 h incubation at 20 °C they were exposed to 35 °C for varying lengths of time and inspected immediately after heat stress. Animals failed to respond to pocking with a pick were scored as dead.

For thermotolerance after the hormetic heat stress, day one adult animals were incubated at 35 °C for 1 h, followed by a 12 h recovery at 20 °C before exposure to 35 °C for 8 h. Worms that displayed egg-laying defect or crawled onto the side of the plate and desiccated were censored and omitted from the analysis.

### Cadmium (Cd²⁺) toxicity assay
Cd$^{2+}$ toxicity assay was performed at 20 °C as described in ref. 3. To estimate survival rate, ~20–25 animals were raised at 20 °C until day 1 of adult and transferred to K-medium (52 mM NaCl and 32 mM KCl) containing 0.25 mM CdCl$_2$ (Sigma-Aldrich) and OP50 in a 12-well plate. The worms were pipetted into new K-medium containing the same CdCl$_2$ concentration every 2 days. For hormetic heat stress, day one adult animals were incubated at 35 °C for 1 h, followed by a 4 h recovery at 20 °C prior to exposure to CdCl$_2$. Dead worms were removed daily.

### RNA-seq and data processing
Stage matched day 1 adult animals that had been raised at 20 °C were used for total RNA isolation with RNeasy Mini Kit (QIAGEN) (strain preparation sees Supplementary Fig. 3a). For *endu-2(tm4977)* strain, we used the fourth generation (G4) of descendance of BR7295 *endu-2(tm4977);byEx1375[endu-2p::endu-2::EGFP;myo-2p::mCherry]* that had lost *endu-2::EGFP* rescue transgene. Purification of poly-A containing RNA molecules, RNA fragmentation, strand-specific random primed cDNA library preparation and 150 bp paired-end reads were carried out on an Illumina HiSeq 4000 by Eurofins Genomics. The RNA-seq results from two independent biological replicates of each experimental condition were uploaded to the European Galaxy Server (https://usegalaxy.eu). All data procedures, including read quality controls, trimming, mapping, and counting, were performed through Galaxy tools[33]. Specifically, we mapped the reads to the *C. elegans* genome using the reference genome (Caenorhabditis_elegans.WBcel235.dna.toplevel.fa) and annotated genes using the reference (Caenorhabditis_elegans.WBcel235.96.gtf) with the RNA aligner STAR 2.6.0b[34]. The reads mapped to individual genes were counted using Feature Counts 1.6.2[35]. For a read to be counted as mapping to a particular gene, we required a minimum read mapping quality of 12 (-Q 12 option of

feature Counts) and an overlap of at least 1 base between the read and any of the exons of the gene (-minOverlap 1). The DESeq2 (Galaxy Version 2.11.40.6 + galaxy1) was used to determine differentially expressed features from count tables of differential transcript abundances[36]. The ENDU-2 dependent post-HS responsive genes were defined with $\log_2 FC(\text{post-HS } vs. \text{ control})_{WT} > 1$, FDR < 0.01 and $\log_2 FC(\text{post-HS } vs. \text{ control})_{WT} - \log_2 FC(\text{post-HS } vs. \text{ control})_{endu-2} > 1$.

## GO term enrichment analysis

GO term analysis was carried out with the online enrichment analysis tool (https://wormbase.org/tools/enrichment/tea/tea.cgi)[37,38]. Then significantly enriched GO terms in the given gene set comparing to the genome background are defined by hypergeometric test. The calcu-

lating formula of P-value is: $P = 1 - \sum_{i=0}^{m-1} \frac{\binom{M}{i}\binom{N-M}{n-i}}{\binom{N}{n}}$. Here $N$ is

the number of all genes with GO annotation; $n$ is the number of genes in the given gene set in $N$; $M$ is the number of all genes that are annotated to a certain GO term; $m$ is the number of genes in the given gene set in $M$. The calculated P-value is then gone through FDR Correction, taking FDR ≤ 0.05 as a threshold. GO terms meeting this condition are defined as significantly enriched GO terms in the gene set. The Graph was generated by OmicShare tool (https://www.omicshare.com).

## RT-qPCR

Total RNA was isolated from ~500 day 1 adult hermaphrodites raised on *E. coli* OP50 bacteria maintained at indicated conditions. RNA was extracted and purified with a RNeasy Mini kit (QIAGEN), and subjected to an additional DNA digestion step. 1 μg DNase-treated total RNA was used as a template for cDNA synthesis using anchored-oligo (dt)$_{18}$ primer and M-MLV reverse transcriptase (Roche). qPCR was performed using Luna Universal master mix (New England Biolabs) in an LC480 LightCycler (Roche). A standard curve was obtained for each primer set by serially diluting a mixture of different complementary DNAs and the standard curves were used to convert the observed CT values to relative values. mRNA levels of target genes were normalized to the mean of the housekeeping gene *act-4*. Primers used for qPCR are listed in Supplementary Data 7.

## Fluorescence microscopy to quantify different expression of different GFP and mCherry reporters

For HS-induced *hsp-16.2p::GFP* expression, day one adult animals were incubated at 35 °C for 4 h before fluorescence microscopy. For expression of *irg-2p::mCherry* reporters after HS, day one adult animals were incubated at 35 °C for 1 h followed by indicated duration of recovery at 20 °C, respectively. Note: As translation and folding of the respective fluorescent reporter proteins occur slowly, fluorescence levels were measured at later time points compared to that for quantitating mRNA levels. For quantification of fluorescence intensities, micrographs were taken from sodium azide-immobilized animals with an Axioimager. Z1 compound microscope using an AxioCam MRm3 CCD camera. Fluorescence of individual worms was captured with the same exposure time and quantified using Fiji[39]. Relative fluorescence to the respective controls was calculated and fluorescence data were pooled across independent experiments.

## Cytoplasmic and nuclear fractionation

Subcellular fractionation was performed as described in ref. 40. Briefly, one-day-old animals grown at 20 °C were washed with M9 buffer until the supernatant was clear and then washed twice with 1 ml cold hypotonic buffer (15 mM HEPES KOH pH7.6, 10 mM KCl, 5 mM MgCl$_2$, 0.1 mM EDTA, 350 mM sucrose). After removal of the hypotonic buffer, the same volume of fresh complete hypotonic buffer (add 1 M DTT and

25× protease inhibitors to the hypotonic buffer to get final concentration of 1 mM DTT and 2× protease inhibitors) was added to the worm pellet. Worms were homogenized with a homogenizer (Slient-Crusher S) with a speed of 75,000 rpm on ice, and the worm debris was removed by twice centrifuge at 500 × g at 4 °C for 5 min. 50 μl of the supernatant was saved as input. The nuclei were pelleted at 4000 × g at 4 °C for 5 min. The supernatant was centrifuged at 17,000 × g for 30 min and the supernatant after this step was "cytoplasmic fraction". The nuclei pellet was washed twice with 500 μl complete hypertonic buffer (15 mM HEPES KOH pH7.6, 400 mM KCl, 5 mM MgCl$_2$, 0.1 mM EDTA, 0.1% Tween 20, 10% Glycerol, add 1 M DTT and 25× protease inhibitors to the hypotonic buffer to get a final concentration of 1 mM DTT and 2× protease inhibitors before use) followed by spinning with 4000 × g at 4 °C for 5 min. The supernatant was discarded and the wash and spin steps were repeated. The supernatant was removed and the pellet was dissolved in a small (50–80 μl) volume of complete hypertonic buffer. The suspension was transferred to a new 1.5 ml tube and this was considered as the 'nuclear fraction'. Note: It is important to change tubes after each wash, as tubulin and fatty substances tend to stick to the tube walls, which may contaminate the nuclear fraction.

## Immunostaining and analysis

Animals were dissected in M9 buffer containing 0.25 mM levamisole on a subbed slide. Samples were immediately fixed in 1% paraformaldehyde in PBS for 10 min. Fixed tissues were incubated in PBSTx (PBS, 0.1% Triton-X-100 and 0.5% BSA) for 5 min to increase permeability. Blocking was performed for 30 min using PBSB (PBS and 0.5% BSA). Primary antibody was incubated overnight at 4 °C in PBSB. The secondary antibody was incubated for 1 h at room temperature. DNA was stained with DAPI with 2 μg/ml for 30 min. Images of intestines were acquired on a Zeiss LSM 710 confocal microscopy. The primary GFP antibody to detect ENDU-2::EGFP in *endu-2(by190[endu-2::EGFP])* day 1 adult animals was a monoclonal mouse anti-GFP antibody (Roche, Nr. 11814460001) and anti-Histone H3 (Abcam) at a dilution of 1:100, the secondary antibody was a goat anti-mouse Alexa Fluor 555 (Invitrogen, A-21422) antibody at a dilution of 1:50. The colocalization degrees between GFP stained ENDU-2 and DAPI-stained DNA has been calculated using Mander's colocalization method[41] performed on Fiji ImageJ software[39].

## High-performance liquid chromatography and mass spectrometry (HPLC-MS)

HPLC-MS analysis was performed on an Ultimate™ 3000 RSLCnano system coupled to an Q Exactive Plus MS instrument (both Thermo Fisher Scientific, Bremen, Germany). For the chromatography a binary solvent system was used with solvent A consisting of 0.1% formic acid and solvent B consisting of 86% acetonitrile and 0.1% formic acid. The HPLC was equipped with two μPAC™ C18 trapping columns (Pharma Fluidics) and a μPAC™ analytical column (length: 500 mm, Pharma Fluidics). Samples were washed and concentrated for 3 min with 0.1% trifluoroacetic acid on the trapping column at a flow rate of 10 μl/min before switching the column in line with the analytical column. A flow rate of 0.30 μl/min was applied to the analytical column and the following gradient was used for peptide separation: 1% B to 25% B in 22 min, to 44% B in 11 min, to 90% B in 2 min, 90% B for 4 min and decreased to 1% B in 0.1 min. The column was re-equilibrated for 18 min with 1% B at a flow rate of 0.40 μl/min. The MS instrument was operated with the following parameters: 1.6 kV spray voltage; 250 °C capillary temperature; for the survey scan: mass range m/z 375 to 1700; resolution at $m/z$ 400 was 70,000; automatic gain control 3 × 10⁶ ions with max. fill time, 60 msec. A TOP12 data-dependent acquisition method was applied with automatic gain control at 1 × 10⁵ ions and max. fill time of 120 ms for high energy collision-induced dissociation of multiply charged peptide ions. The normalized collision energy (NCE) was set to 28%. Dynamic exclusion was set to 45 s.

## In-gel protein digestion

Co-immunoprecipitation eluates were loaded onto NuPAGE® Novex® 4–12% Bis-Tris gels (Invitrogen, Life Technologies). After gel electrophoresis, proteins were stained with Coomassie Brilliant Blue G-250, and gel lanes were cut into approx. ten slices of equal size. Slices were washed and destained by alternatingly incubating them with 10 mM ammonium bicarbonate and 50% ethanol/10 mM ammonium bicarbonate (10 min at RT each). For reduction of disulfide bonds and subsequent alkylation, 5 mM tris(2-carboxyethyl) phosphine (10 min at 60 °C) and 100 mM 2-chloroacetamide (15 min at 37 °C) were used, respectively. Proteins were digested with trypsin (Promega, Mannheim, Germany) in 10 mM ammonium bicarbonate at 37 °C overnight and peptides were extracted using one volume of 0.1% trifluoroacetic acid in water mixed with one volume of 100% acetonitrile. The peptides were dried down and taken up in 15 μl 0.1% trifluoroacetic acid in water.

## MS data analysis

Rawfiles were searched with MaxQuant version 2.0.2.0[42,43] against the *C. elegans* UniProt reference proteome (ID: UP000001940; Taxonomy: 6239; release: 2022_01; 26584 protein entries). Default settings were used in MaxQuant, with Trypsin/P as proteolytic enzyme and up to two missed cleavages allowed. 1% false discovery rate was applied on both peptide and protein level. Oxidation of methionine and N-terminal acetylation were set as variable and carbamidomethylation of cysteine as fixed modifications. Label-free quantification[44] was enabled, with a minimum ratio count of 2. For data analysis, the proteingroups.txt file of MaxQuant was used and loaded into Perseus 1.6.15.0[45]. Entries for reverse and potential contaminant hits as well as proteins only identified by site were excluded from the analysis. LFQ intensities were log$_2$-transformed. Only protein groups with LFQ intensities in all replicates of the ENDU-2-EGFP IP or in all replicates of the corresponding control IPs were considered for further analysis. Missing values were imputed from normal distribution using the following settings in Perseus: width 0.5 and down shift 1.8. The protein groups were further filtered for at least to unique peptides identified and a sequence coverage >5 %.

## Co-immunoprecipitation (co-IP) and western blot

Day 1 adult animals were lysed in cold NP-40 buffer (150 mM NaCl, 50 mM Tris HCl pH 8.0, 0.1% NP-40) supplemented with cocktails of protease inhibitors (Sigma-Aldrich) and then treated with RNase A (Thermo Scientific) or RNase inhibitor (Thermo Scientific) for 30 min at 37 °C, and sonicated for 30 sec twice using SlientCrusher at 75,000 rpm. Protein extracts were centrifuged at 12,000 × g at 4 °C for 20 min. After quantification of protein concentration with the Bradford assay, 1 mg of protein extract was incubated with 2 μg of anti-Pol II 8WG16 (AMA-1) (Santa Cruz Biotechnology, sc-56767), 2 μg anti-POLR2B (RBP-2) (Invitrogen, PA5-30122), 1 μg anti-Histone H3 (Abcam, ab1791) or 1 μg anti-GFP (Abcam, ab290) for 1 h at 4 °C. The immune complexes were then incubated with 30 μl Dynabeads Protein A (Invitrogen) for overnight at 4 °C. After four washes with the wash buffer (PBS, 0.5% Triton X-100, 1 mM EDTA) the beads were resuspended in 30 μl of 1× Laemmli Sample buffer and incubated at 95 °C for 5 min. Total protein or co-immunoprecipitated samples were resolved by 10% SDS-PAGE and transferred to PVDF membrane (Bio-Rad). The membrane was blocked with 5% non-fat milk for 2 h at room temperature, incubated with primary antibody (1:2000 dilutions of anti-GFP, anti-Pol II 8WG16, anti-POLR2B or anti-Histone H3) overnight at 4 °C, and after wash 3 times, the blot was incubated with HRP-conjugated secondary antibodies (Invitrogen, 31460 and 31430, 1:5000) for 1 h at room temperature. Immunoblot signals were detected by LAS-4000 (Fujifilm). For ENDU-2 MS interactome study, *endu-2(by190[endu-2::EGFP])* of day one adulthood stages were incubated at 35˚C for 1 h and then collected for co-IP. As wash-off the

animals took about 2 h at room temperature, we considered these animals were recovered from HS for 2 h.

## Analyzing RNA quality

The RNA quality was assessed using gel electrophoresis on an 1% TAE buffer-based agarose gels containing 1% Sodium Hypochlorite (Carl Roth)[46]. Each gel was loaded with 10 μL 1× DNA Loading Buffer containing 2 μg of total RNA isolated from protein samples for co-IP studies and run for 35 min at a constant voltage of 100 V.

## Chromatin immunoprecipitation

Chromatin Immunoprecipitation (ChIP) experiments were performed as described in ref. [47], [48]. For Pol II ChIP, WT and *endu-2(tm4977)*, for ENDU-2 ChIP *endu-2(by190[endu-2::EGFP])* day 1 adult animals were used. Worms were washed with M9 buffer, and then cross-linked with 2 % formaldehyde for 30 mins at room temperature followed by quenching with 0.4 ml 2.5 M glycine for 5 mins at room temperature. For *endu-2(by190[endu-2::EGFP])* worms, an additional treatment of 10 mM dimethyl 3, 3'-dithiobispropionimidate (DTBP, Thermo Fisher Scientific) followed by quenching with 2.5 ml 2.5 M glycine for 5 mins before cross-link was needed. Worm were the washed with M9 and B-ChIP-L0 (with protease inhibitors and PMSF) buffer. Worm pellets were resuspended in 0.8 ml B-ChIP-Lys (with protease inhibitors and PMSF) buffer, followed by homogenization and sonication (20 × 10 sec. on, 59.9 sec. off, with a digital Branson sonifier) in a volume of 2 ml. Lysates were spun at 12,000 × g for 15 mins at 4 °C. The supernatant was removed and the protein concentration was determined by Bradford method and 4 mg and 12 mg extracts were used for RNA Polymerase II ChIP and ENDU-2::EGFP ChIP respectively. 10 % of each IP sample was taken as input. 4 μg of anti-Pol II 8WG16 (AMA-1) (Santa Cruz Biotechnology) or 1 μg anti-GFP (Abcam) was added to the IP sample and then incubated overnight at 4 °C. Immune complexes were recovered using Dynabeads Protein A (Invitrogen) and washed at 4 °C with 1 ml of each of the flowing solutions: B-ChIP-Lys (2 × 5 mins), B-ChIP-Lys (200 mM NaCl) (1 × 5 mins), B-ChIP-W (2 × 5 mins), and TE (50 mM NaCl) buffer (2 × 5 mins). Samples were eluted with 300 μl B-ChIP-EL elution buffer for 15 mins at 65 °C. DNA were purified using RNase A plus proteinase K followed by an overnight incubation at 65 °C. Cleaning and concentrating each input and IP samples using a ChIP DNA clean up kit (Biozol). qPCR using Luna Universal master mix (New England Biolabs) was performed on inputs and ChIP samples using the primer sets listed in Supplementary Data. 7. Three independent, biological replicates were performed. Fold enrichment was calculated using the ΔΔCt method.

## Statistical analysis

Statistical analysis was performed on the GraphPad Prism 9.3.1 (471) software, except for RNA-seq and MS analysis, which was performed as described above. Experiments shown in this study were performed independently for 2 to 5 times. Details of the particular statistical analyses used, precise *P* values, statistical significance, number of biological replicas, and sample sizes for all of the graphs are indicated in figures or figure legends. Unless mentioned otherwise. *n* represents the number of animals tested, *N* is the number of biological replicates.

## Reporting summary

Further information on research design is available in the Nature Portfolio Reporting Summary linked to this article.

## Data availability

The RNA-seq data are deposited in BioProject under BioProject ID PRJNA881926. The results of RNA-seq data are located in Supplementary Data 1–3. The proteomics data are available via ProteomeXchange with identifier PXD041872. Source data are provided with this paper.

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

## Acknowledgements

We thank the Caenorhabditis elegans Genetics Center, which is funded by the NIH Office of Research Infrastructure Programs (P40 OD010440) for providing some strains; The ProteomeXchange Consortium to provide globally coordinated standard data submission. Dr. S. Mitani and National BioResource Project (NBRP) for providing *endu-2(tm4977)* strain. We thank the staff of the Life Imaging Center (LIC) of the University of Freiburg for their microscopy resources, Spemann Graduate School of Biology and Medicine for providing training courses. We thank China Scholarship Council (CSC) for providing financial support to our co-author R.L. We thank Dr. Nicola Iovino for the establishment of ENDU-2 ChIP, Dr. Ritwick Sawarkar, professor Dr. Thomas Laux and Dr. Cecere Germano for discussion. This work was funded by grants from the German Research Foundation (DFG) (SFB1381), Germany's Excellence strategy (CIBSS-EXC-2189 - Project ID 8 390939984) to R.B. and B.W.

## Author contributions

F.X. and W.Q. designed the experiments. F.X. performed the majority of the experiments. R.L. performed some immunofluorescence experiments. E.D.G performed cell fractionation experiments; F.X., E.D.G, F.D., B.K., and B.W. contributed to mass spectrometry experiments and data analysis; F.X., W.J.Q., and R.B. wrote the manuscript, R.B., and B.W. provided funding for the project and conceptional suggestions to its execution.

## Funding

## Competing interests

The authors declare no competing interests.
