## [Peer Review File · Nature Communications]

Reprogramming of the transcriptome after heat stress mediates heat hormesis in *Caenorhabditis elegans*REVIEWER COMMENTS

Reviewer #1 (Remarks to the Author):

This work tests the hypothesis that *C. elegans* *endu-2* is specifically involved in regulation of gene expression after (but not during) heat shock, and explores some of the potential mechanistic ways it might achieve this.

The authors build on earlier work focusing on ENDU-2, an RNA-binding protein and a polyU-specific endoribonuclease of the model organism *C. elegans*. ENDU-2 “responds to” (=?) “adverse” (=?) temperatures and the authors previously showed it is a secreted protein, which motivates them to look at its possible involvement in heat shock (-> not involved) and heat hormesis (-> partially involved) responses.

The work is interesting because it unveils some of the appreciable complexity of this molecule, showing it can act at different subcellular locations, and is relevant to hormetic, but not immediate post-stressor effects in the experimental context. It is refreshing to see researchers embrace the multifaceted reality this protein seems to act in, explore several options, deliver valuable new insights and open up exciting avenues for further research. I have no doubts that the large number of experiments were executed rigorously and held to high standards. I do have important questions on experimental and interpretational logic, as explained below.

Major (relevant details included in text further below):

The narrative (mainly in the introduction, to a lesser extent later) does not always flow logically, and everything makes sense only when the entire work is read. Some questions/suggestions for improvement are given in the ‘detailed feedback’ below.

This work follows a number of sensible leads, but also makes other claims without experimentally sorting them out. A major aspect leaving some confusion here is the reasoning associated with cell-autonomous vs non-autonomous actions, tissue specificity and subcellular localization, which are not resolved based on the data presented. This should not be hugely problematic, but the work should discuss this far better (verify what can be stated throughout, and then discussion section can be extended).

With their “class I-II-III” genes, the authors seem to “re-invent” early-intermediate-late responses to stressors, while not mentioning those concepts at all. To discuss discriminating acute and hormetic responses, this is nonetheless relevant. Could they place their observations in the context of these concepts, and perhaps explain why they refrain from such terminology (for now)?

There is a major problem with causality that needs to be resolved (see detailed feedback below). While it is beyond doubt that ENDU-2 responds to heat stress in a hormesis-relevant way, for some of the core experiments, causality has not been sorted out sufficiently.

1. Hormetic lifespan of control, mutant and rescues needs to be sorted out (this is essential)
2. The RNAseq data depend on the assumption that the full effect is due to loss of *endu-2*, but the ultimate proof of causality needed to support that (see 1.) is lacking for now
3. nested effects of time need to be addressed in transcriptomics data

Where are the omics data (transcriptomics and proteomics) deposited? These data do not seem available.

Detailed feedback:

There are too many language errors scattered throughout the text, and also several overstatements or (sub)sentences with unclear relevance used. E.g. line 4 “Any” organism is “constantly” exposed to ... , or lines 53-54 which refer to research on mice, but are phrased as if human-based. Please don’t focus on these two examples only during revision; there are

several more sentences in need of attention in the text; especially the introduction is sensitive to this, but also elsewhere. I would advise to scrutinize the text so that phrasing may be nuanced, language errors corrected, and the reader will be able to focus on what matters: the findings and interpretations. As a minor comment on the side: the authors regularly indicate whether they believe certain topics are researched more/less; this is unnecessary and I would propose to slim the text down to the scientific narrative only.

Lines 34-46 Are these orthologs, or simply examples of ENDOU family members? How are the examples given (viral, human proteins) relevant for the topic and ENDOU-2 focus in this work? *C. elegans* seems to have two ENDOU proteins, I would find it more informative to use the introduction to help the reader understand their relation to other ENDOU family members and to heat (and cold – unless there are good reasons to assume that is mechanistically different here) stress, than to see these examples.

Line 45 If ENDOU-2 is a secreted protein, does it have cell-autonomous actions? Apart from the author's previous work, do we know whether any of the functional implications are due to cell-(non)-autonomous actions? If both exist, could the introduction clarify this?

Line 61 The logic/narrative of the current introduction appears a bit random. Even when revisiting it after having read everything, it still did not fit smoothly with what I summarized out of the entire work: "This work tests the hypothesis that *C. elegans* *endu-2* is specifically involved in regulation of gene expression after (but not during) heat shock, and explores some of the potential mechanistic ways it might achieve this."

The authors mention a broader goal: to understand (assumedly via an example study on ENDOU-2?) how RBPs may affect transcription through co-transcriptional RNA processing (cf lines 30-33?), but that is not exactly what is being tested in this work. If this is the overall goal, then why ENDOU-2 and no other RBP? Why both acute and hormetic treatments, how is that relevant for that conceptual question? ... Or is the goal to understand differences between acute HS response and long-term hormetic effects in an ectothermic context? If so, then why the focus on ENDOU-2? Also, in this scenario, starting from a current view on hormetic lifespan effects and known regulatory mechanisms may be interesting? In any case, I am a bit confused by the motivation for the experiments done here, which the authors can probably easily address by adjustments to their introduction.

It is stated that strains were maintained at 20°C, however, the authors previously reported that at this temperature, there is progressive loss of brood size over generations in *endu-2* loss of function mutants, ultimately leading to sterility (temperature-dependent Mrt phenotype). Aren't the *endu-2* mutants cultured at 15, then, which is when they do not have any such problems? When were they changed to 20 °C? If this happened, was this also done for all other experimental strains, so that there are no temperature confounders in the observations? How many generations prior to executing experiments on them? ... wait, this is probably 'explained' in lines 625-626. So the question would be: are the first five generations fine? This paper should stand sufficiently on its own, so I would appreciate a more extensive description in the methods here, rather than an off-hand reference to previous work, for something that is quite relevant to the current study. For example, how did they deal with the tissue-specific rescue strains, then? Are we dealing with other temperature histories of different strains? After how many generations is an epigenetic effect of temperature negligible? Please explain.

The authors should specify what "day 1 of adulthood" precisely is, and how they generated synchronized populations. Nearly all experiments rely on (begin with) day 1 adults, but without defining this, the manuscript does not meet the reporting requirements.

Fig 1 & lines 90-96: there are a number of non-trivial considerations to be made with these lifespan experiments.

For one, it is unclear how the LogRank p values shown for panels b-i (all survival curves) were derived, and how this analysis took replicate experiments into account. Some of the p values seem odd; for example, for panel h, 0.5953 is a weirdly low p value, when looking at the survival curves. This could be due to showing pooled data, while properly recognizing independent experiments in the statistical analysis, but it needs to be sorted out, because it is impossible to assess at the moment. Survival data are variable (e.g. already easily visible in this figure if one imagines all control conditions that are presumedly effectless in the same graph); that is by no means the authors' doing, but a well-recognized reality of the readout. The authors however are faced with combining this reality with only small (if any) effects observed in panels e-i; therefore showing the replicate experiments (e.g. in supplement), rather than only pooled data, as well as clarifying the statistics are a must.

In addition, one very worrying fact that is left unaddressed, is that the endogenous rescue does not fully rescue. This may mean the effect seen in panel b is actually not due to *endu-2*, but rather due to a background mutation or an epistatic effect. It may alternatively mean that the construct they used for rescue, a fusion protein with GFP, is unable to do so. What are the results with a proper endogenous rescue (e.g. using a trans-splicing sequence to avoid the GFP being attached to *ENDU-2*, or even omitting it altogether)? If these are different, then also all tissue-specific rescues will need to be redone. If the results are similar to those of panel e, then what to make of this conceptually, in comparison with the effect observed in panel b? When testing the endogenous rescue, I would advise to also include control conditions in the same experiments and graphs – how much of the wt vs *endu-2(tm4977)* effect seen in these strains is really due to *endu-2*?

Then, is it possible to consider effect sizes also in the tissue specificity? For example, the muscle rescue looks quite similar to the endogenous one. Given this is a secreted protein, perhaps the source of production is less relevant to its ultimate effect on longevity, then the available dose is. Not all tissues may produce it at similar levels, and the tissue-specific promoters used likely do not replicate this faithfully? In fact, conditional knockouts may be more interesting to address the question of tissue-dependent contributions to the phenotype, here?

Lines 95-96: Related to the above: it is stated *ENDU-2* is a secreted protein. Would it not be equally likely that these tissues are not the main ones “requiring” it, but happen to be the main ones producing it, depending on the strength of the promoters used?

Line 143 What is the motivation for deciding on these time points (mainly a question for the 4h one, obviously)? This may also be a good place to already insert the notion elaborated further on in lines 580-586, that this is a dynamic process and temporal choices have to be made to study its dynamics from an omics perspective.

Minor: while it is common to use up- and downregulated in omics analyses, the terminology is suggestive in a slightly misleading way. The authors quantified RNA levels, that can be higher/lower or in-/decreased in one condition vs another, but that is a result of more than merely up- or downregulation of transcription. I would suggest to correct this throughout the entire document, or, if that is too cumbersome, to at least very explicitly state what is meant with the terms in the manuscript.

Lines 152-155 (and paragraph below, and later on as well) Yes and no. One concern is that there is not a temporally matched, non-heat shocked control. *C. elegans* transcriptomes are known to be dynamic in early adulthood, so a non-heat shocked population that is 6 hours older than its reference will also display a number of significant transcriptomic differences as a result of that. In the experimental setup used here, these effects are nested with those of the late (incl. hormetic) response. Could the authors explain why they did not include such a control, and adjust their interpretations wherever needed in this manuscript, to take into account that a post-HS animal differs from controls in more ways than only by having undergone the treatment?

Line 176 what were the selection criteria for these candidates to confirm? How many of these were expected to match expectations, how many were not/true unknowns?

What happens with endu-2 RNA during and after HS? Would that not be relevant to highlight in the discussion of the results as well?

Lines 181-189 (and more general) ENDU-2 is necessary for hormetic effects after HS (standard lifespan and Cd²⁺ survival alike). This means ENDU-2 needs to “know” that a HS has happened (on and off again). Is ENDU-2 a directly heat sensitive protein (cf also nice observations in lines 521-526)? Can hsf-1 mutants still raise the endu-2 response? The latter seems straightforward to test, and would be interesting to strengthen the message of this manuscript? (The same could be asked for daf-16, daf-12 and swsn-1 based on literature given as inspiration in lines 272-273, but hsf-1 seems most relevant for this question?)

Lines 190-194 I agree that is what is suggested, but how to make sense of this? The authors state they previously showed this protein is targeted to the secretory pathway. How do they think it then acts cell-autonomously and/or moves between different subcellular compartments (see also the very nice experiments showing it can do so, but I miss an interpretation that brings together those bits of information)?

Figure 2d can the heat map be provided with all its labels in the extended data? None of the rows are labelled, but this is valuable information. It is impossible to include this in the main figure, but of scientific value to add it to the manuscript.

Extended Data Figure 3

(a) It is unclear to me what the tissue enrichment representation is based on, and the methods section on this (666-669) does not clarify it. What data source(s) feed into this? How many tissues are linked to a transcript? What are the criteria? What does the size of the bars (the “-log₁₀ q value”) mean? I have the impression this was glanced over a bit too superficially, because it matches the expectations. This may remain true after explanation, but as it is, the data shown are not particularly valuable to me.

(b) is there a table containing the detailed information of this available as well? Please refer to where it can be found? Without those data, these are just some numbers in a Venn diagram.

Lines 272-283: how do the authors fit SWSN-1 and ENDU-2 together in their model of heat response/hormesis? This feels incomplete and somewhat out of place in the flow of the manuscript. Why bother testing this if not to connect it to their protein under study? They have an ENDU-2 reporter strain; did they look at it signal after SWSN-1 RNAi? Do we have two proteins affecting irg-2 transcription, or is one under control of the other? Also, in this paragraph, can the authors please explicitly clarify that RNAi in their paradigm does not target the nervous system? This is relevant in the context of their cell-(non-)autonomous logic that is always at the back of one’s mind when reading this manuscript.

Lines 337-366 are a nice exploration, even though a bit of a lucky shot with then a negative result. Is ZIP-2 under control of SWSN-1? Together with the previous remark, this may tie up the narrative a bit more? Does *C. elegans* have DDIT3 homologs? Might be worth mentioning in this narrative?

Lines 353-360 Transcriptomics does not seem to rely on full-length reads (cf methods). Then why is it permitted to compare read counts in this part of the manuscript?

Line 390 and onward and figure 9: this is very nice work, but how to make sense of this for a

secretory pathway protein (see above)? Are SWSN-1 or ZIP-2 amongst the proteomic partners? Are there any other interesting partners in the list? Now, it seems, the only thing the MS experiments support, is the lead that ENDU-2 might also exist in the nucleus. -> line 488 provides RPB-2 as “another” hit, but there are no clear hits mentioned before this (unless I missed that?)?

Lines 439-440 Yes. Or that these amino acids are needed for something else as well? Can that be ruled out or considered unlikely? If so, please mention why?

Line 490 & extended data 8: AMA-1 was not enriched in the ENDU-2 pulldown MS, why was it chosen for co-IP? Were these the only Pol II proteins that were identified by MS? If so, then please mention, but also, there is no real predictive value in being a “better” hit from the MS analysis? If not, could others please also be indicated in Extended Data Figure 8, and could it be explained why those were not focused on?

Reviewer #2 (Remarks to the Author):

This paper reports a key role for the endoribonuclease ENDU-2 in transcriptional modulation of the benefits of hormesis. A well-established model of heat shock hormesis is used to examine the mechanisms of ENDU-2. The authors find that ENDU-2 is required for hormetic lifespan extension as well as hormetic resistance to cadmium. The effects of ENDU-2 are reported to require ENDU-2 association with chromatin upon heat shock; recruitment of RNA Pol II; and results in transcription of a distinct set of genes, some of which (“Class III”) occurs after the acute heat shock response has subsided. This post-heat-shock adaptation can be decoupled from the acute heat shock response.

This is a data-rich and appropriately-controlled study in which the (surprising?) conclusion that EDNU-2 acts as a nuclear transcription factor is well-supported. There are a few minor issues to address:

1) The manuscript does not cite Jia et al, who claimed that the germline effects of EDNU-2 in response to genotoxic stress were mediated largely through CTPS-1 phosphorylation, controlled by down-regulation of PKA and HDA-1. While this may be unrelated to the hormetic effects of EDNU-2, it would be interesting to know if the transcripts of any of these genes were altered in the investigator’s transcriptome data.

2) lines 439, 440 - The claim that both RNA-binding and cleavage are essential to mediate the beneficial effect of heat hormesis is not the logical conclusion of the data presented. Since the E460Q mutation affects both RNA binding and RNA cleavage, it remains possible that only RNA cleavage is required for the hormetic effect.

3) The probability values in extended data Fig 5a appear to be switched.

4) Given the proposed role of EDNU-2 as a transcription factor, it would have been informative to have done a ChIP-seq experiment with the EDNU-2::GFP strain, as this could have identified all the genes under EDNU-2 control, and potentially revealed any sequence motif responsible for EDNU-2 chromatin binding. However, given the extensive analysis already presented in this manuscript, this is not something that needs to be added to this study.

Reviewer #3 (Remarks to the Author):

In this manuscript the authors investigate how the benefits of hormetic heat stress are manifested over time on a transcriptional level. This is an interesting, well-executed manuscript that addresses an important question in aging biology and stress-response hormesis. Specific findings and comments under each section heading.

ENDU-2 contributes to hormetic heat stress-mediated beneficial effects

The authors show that *endu-2* mutants have no defect in resistance to heat stress but that *endu-2* is required for CdCl₂ resistance and hormetic HS-mediated lifespan extension. The *endu-2* rescue experiments are not convincing and need to be strengthened. Specifically:

- The authors should describe the *endu-2* mutation better and demonstrate or cite previous research confirming loss-of-function.

- The expression pattern of ENDU-2 should be described better and the authors should include a supplementary figure showing the expressing pattern of the ENDU-2::GFP constructs they are using for lifespan.

It would be interesting to determine whether the benefits of hormetic heat shock extend to other stresses besides CdCl₂ resistance, especially the reported increased resistance to heat stress. This would strengthen their findings that ENDU-2 is involved in conveying multiple hormetic benefits and could be used as additional read-outs in subsequent figures, since the authors only examine CdCl₂ resistance and do not show additional lifespan experiments.

- Lifespan experiments: All lifespan/survival assays should be performed at least three times independently. Typically, lifespan data are not pooled but reported as independent experiments and all data listed in tables. The lifespan data in Figure 1e-i need to be performed with WT controls, to demonstrate that the heat shock induced hormetic benefits. The authors did not include in the methods sections what program log-rank test was calculated with. We recommend OASIS2 (a web-based survival calculation program). In Fig 1e to 1i, the levels of ENDU-2 rescue are not clear and need to be addressed. Was the *endu-2* level back to WT? Or they are overexpressing *endu-2*?

Lifespan assays were performed by adding FUdR. Even only they used for the first 7 days, the FUdR still did affect animals. The authors should at least show one set of lifespan without adding FUdR.

- Fig 1a should use two-way AVOVA not multiple t test for statistics.

Hormetic heat stress results in an ENDU-2 dependent reprogramming of the transcriptome after HS

In this section the authors nicely demonstrate the *endu-2* dependent gene transcription changes that occur immediately after heat shock and compare them to transcriptional changes occurring after the acute heat shock with 4h recovery. We have some suggestions to improve the clarity of this section.

- Line 184-185, more information about the function of these genes chosen as representative for each class would be helpful, especially in the case of *irg-2*, which is further explored.

- Line 190-191, "...genes was enriched in the intestine and muscle (extended data fig 3a)." The authors should explain this better, are they known intestinal and muscle-specific genes? This is important to clarify since the others did bulk RNASeq and not did not isolate tissues.

- Figure 2d would benefit from a schematic of the classes as presented in Fig 7, or show the pattern of the selected genes representative for each class over time

- In Fig 2f, it seems that all selected genes are increased in *endu-2(lf)* at 20°C. What are the statistics of these genes for WT and *endu-2(lf)* at 20 °C? The authors should include a comparison of their RNAseq data of WT with *endu-2* mutants under basal conditions. How many genes are differentially regulated in the mutants and do these genes fall into the HS-regulated classes? The authors should use two-way ANOVA for their statistical tests.

- Extended Data Fig2: The authors need to include an additional repeat and should include either Lifespans or another hormetic benefit in these analyses. Why would these specific

genes be required for CdCl₂ resistance? What is the physiological function and relevance here?

ENDU-2 positively regulates gene expression at the transcriptional level after HS

The authors postulate that ENDU-2, a mRNA-binding endoribonuclease regulates gene expression on a transcriptional level and they are using transcriptional reporter constructs to differentiate between transcriptional and post-transcriptional regulation and compare Class I and Class II genes. These experiments do not answer this question.

- Since Fluorescence readouts require the synthesis of a fluorescent protein and are thus not direct read-outs for transcription.

- The experimental set-up for class I and Class II fluorescent reporters is different. TJ375 worms will be fluorescent after a 1h HS and 2h Recovery, but not after just 1h HS which was reported in their RNASeq experiment, most likely due to a delay in protein synthesis of GFP. To determine the different transcriptional class the reporter strains need to be imaged at the same HS and recovery times. Moreover in line 264 the authors state the *irg-2* reporter was measured after 8h while the figure legend states 16h.

- In line 267, the authors state "Surprisingly, hormetic HS failed to activate *irg-2p::mCherry*". Why is this surprising, they already showed this by RNASeq and qRT-PCR?

- In extended Data Fig 5A, the p-values seem to be swapped.

Activation of the bZIP transcriptional factor ZIP-2 in the post-HS phase is independent of ENDU-2

The authors hypothesized that ZIP-2 (responsible for regulating *irg-2* gene expression) is activated by ENDU-2 and that ZIP-2 would in turn activate the post-HS gene expression, they find a possible activation of ZIP-2 which is however independent of ENDU-2.

- This section seems unnecessary and distracts from the main finding of this manuscript. Should the authors wish to keep this section in the manuscript they should address whether ZIP-2 is required for the hormetic benefits similarly to *irg-2*.

- The images of nuclear localization are fuzzy. The authors need to provide better images if they tried to claim ZIP-2 translocating into nucleus after HS but not in *endu-2(lf)*.

ENDU-2 localizes in the nucleus and associates with chromatin

In this section the authors identified that ENDU-2 is localized in the nucleus and cytoplasmic fraction and can be immunoprecipitated with Histone H3. Nuclear localization of ENDU-2 is increased with HS. This section could be strengthened with additional experiments:

- The ENDU-2 IP followed by Mass spec is not well described in the results or in the methods sections and should be improved. The authors should consider describing the nuclear proteins that ENDU-2 is associated with in more details. Was Histone H3 found in mass spec? Maybe their own data can help strengthen the claim that ENDU-2 is associated with chromatin

- In line 405-406 the RNAase treatment and RNA-independence should be described in more detail as it is not clear. Did the authors have any control for RNase A digestion in Fig 4c? How did they know all the RNA in their reaction was gone?

- Since the authors are capable to isolate nuclear and cytoplasmic fraction, does HS and HS followed by recovery induce more ENDU-2 in nuclear fraction?

1. Fig 4b and 4d, they should use the chromatin marker (e.g. histone) to examine if ENDU-2 colocalized with chromatin, rather than DAPI staining.

Both RNA-binding and -cleavage activities of ENDU-2 are essential for mediating heat hormesis.

The authors use RNA-binding and RNA-cleaving mutants of ENDU-2 to determine their function in heat hormesis and find that while both activities are required for the transcriptional regulation of *irg-2* upon HS. The imaging data is confusing and does not fit with their nuclear cytoplasmic fractionation, which again should be performed +/- HS.

- Line 441/442, if the authors want to identify whether ENDU-2(E454Q) and E460Q are

chromatin associated, they should provide histone H3 IP results and use chromatin marker in extended fig 7 as well. The current data they provided only shows ENDU-2(E454Q) and E460Q exist in nucleus but not chromatin associated

- Line 447, "...incapable to localize on chromatin..." and line 448, "...the chromatin association of ENDU-2..." again, their data did not support these statements.
- An additional readout for hormetic benefits, such as LS or stress resistance should be added

ENDU-2 activates RNA polymerase II to trigger post HS response

Here the authors provide intriguing and nice mechanistic evidence of how ENDU-2 affects the transcriptional response after hormetic HS.

2. In fig 6a and 6d, what are the major bands (a bit smaller than 75 kDa) in the anti-GFP blot when IP AMA-1? They are abundant and even appear in GFP control group.

Minor comments:

1. Why do authors include right Y axis in all survival curve figures? Is it really necessary?
2. Line 41, stresse should be "stresses."
3. The label of the promoter of the gene in transgenes is incorrect. According to *Caenorhabditis* nomenclature 4.5.4. (http://www.wormbook.org/chapters/www_nomenclature/caenornomenclature.html#sec4.5), the lowercase p is used to indicate the promoter of the gene rather than uppercase.
4. In line 113, a p is missing in *myo-2p::mCherry* transgene. It should be *myo-2p::mCherry*.
5. Line 118, mediating "the" beneficial effect of hormetic HS.
6. Line 320, y is missing in *irg-2p::mCherry*.
7. Line 380, a space is lacking between "two independent... "
8. Figure 4b "intestine (20 oC)" e is missing.
9. Merge images for fig 4d are also helpful for readers to see the colocalization even though the authors provide colocalization coefficient.
10. The authors should quantify the colocalization coefficient for extended fig 7c and 7d as well and provide merge images of chromatin marker and ENDU-2.
11. Line 440, to mediate "the" beneficial effect.
12. Line 877, ref 36 "*Caenorhabditis elegans*" is missing after "from" in the title.

Rebuttal letters

Manuscript ID: NCOMMS-22-51291-T

Manuscript title: Reprogramming of transcriptome after hormetic heat stress by the endoribonuclease ENDU-2 improves lifespan and stress resistance in *C. Elegans*

We have addressed all the questions from the 3 reviewers and modified introduction and discussion according to the suggestions. The new results in the revised manuscript are highlighted in yellow.

REVIEWER COMMENTS

Reviewer #1 (Remarks to the Author):

This work tests the hypothesis that *C. elegans* endu-2 is specifically involved in regulation of gene expression after (but not during) heat shock, and explores some of the potential mechanistic ways it might achieve this. The authors build on earlier work focusing on ENDU-2, an RNA-binding protein and a polyU-specific endoribonuclease of the model organism *C. elegans*. ENDU-2 “responds to” (=?) “adverse” (=?) temperatures and the authors previously showed it is a secreted protein, which motivates them to look at its possible involvement in heat shock (-> not involved) and heat hormesis (-> partially involved) responses. The work is interesting because it unveils some of the appreciable complexity of this molecule, showing it can act at different subcellular locations, and is relevant to hormetic, but not immediate post-stressor effects in the experimental context. It is refreshing to see researchers embrace the multifaceted reality this protein seems to act in, explore several options, deliver valuable new insights and open up exciting avenues for further research. I have no doubts that the large number of experiments were executed rigorously and held to high standards. I do have important questions on experimental and interpretational logic, as explained below.

Major (relevant details included in text further below):

The narrative (mainly in the introduction, to a lesser extent later) does not always flow logically, and everything makes sense only when the entire work is read. Some questions/suggestions for improvement are given in the ‘detailed feedback’ below. This work follows a number of sensible leads, but also makes other claims without experimentally sorting them out. A major aspect leaving some confusion here is the reasoning associated with cell-autonomous vs non-autonomous actions, tissue specificity and subcellular localization, which are not resolved based on the data presented. This should not be hugely problematic, but the work should discuss this far better (verify what can be stated throughout, and then discussion section can be extended).

Response: In the revised version we provide additional data supporting an autonomous activity of ENDU-2 (see responses to the relevant details below) and have intensively modified the discussion section to analysis different aspects of ENDU-2 function.

With their “class I-II-III” genes, the authors seem to “re-invent” early-intermediate-late responses to stressors, while not mentioning those concepts at all. To discuss discriminating acute and hormetic responses, this is nonetheless relevant. Could they place their observations in the context of these concepts, and perhaps explain why they refrain from such terminology (for now)?

Response: The reason for using this classification instead of “early-intermediate-late responses “ is to differentiate our study from a previous one which investigated alteration in gene expression in

response to a time series of HS¹. In this study they described genes showing altered expression levels with short HS as “early responsive”, and genes with changed expression levels only upon long-duration of HS as “late responsive”. We also compared our class III genes with those “late responsive genes” to HS and found they belong to a different set of genes (Supplementary Fig. 3e). One good example is that *irg-2* can be only activated after HS (Supplementary Fig. 6). Therefore, we will stay with our terminology but added the explanation in the result part.

There is a major problem with causality that needs to be resolved (see detailed feedback below). While it is beyond doubt that ENDU-2 responds to heat stress in a hormesis-relevant way, for some of the core experiments, causality has not been sorted out sufficiently. Hormetic lifespan of control, mutant and rescues needs to be sorted out (this is essential)

Response: In the revised version, we have included a series of lifespan experiments, including another *endu-2(lf)* allele *by188*, the *endu-2::EGFP* knock-in strain to check influence of EGFP fusion, two additional *endu-2::EGFP* rescue lines that have less strong overexpression of ENDU-2::EGFP. We could show that the second *endu-2(lf)* allele *by188* behaved similarly as the first one *tm4977* in lifespan (Supplementary Fig. 1c). EGFP fusion to ENDU-2 did not interfere with heat hormesis (Supplementary Fig. 1e), but rather excessive ENDU-2::EGFP overexpression (Supplementary Fig. 2b-d).

Moreover, we added an additional assay to access beneficial effect of short HS: thermotolerance upon a subsequent HS (Fig. 1e). Again, hormetic HS resulted in improved thermotolerance of WT but not *endu-2(lf)* mutant animals. In addition, our *endu-2::EGFP* rescue strain under control of its own or intestinal promoter, which failed to completely rescue the lifespan phenotype, completely rescued the thermotolerance phenotype, muscular expression seemed to have a weak effect while neuronal and somatic gonadal expressed *endu-2::EGFP* failed to do so.

All these data together suggest that although ENDU-2 mediates the heat hormesis, too much ENDU-2 protein might have deleterious impact at late stage that eventually interfere with the benefits acquired at earlier stage.

The RNAseq data depend on the assumption that the full effect is due to loss of *endu-2*, but the ultimate proof of causality needed to support that (see 1.) is lacking for now.

We have tested whether *endu-2::EGFP* rescues the expression level of *hsp-16.2* and *irg-2* mRNA with qPCR (Fig. 5d). In both cases, transgenic *endu-2::EGFP* but not *endu-2(E454Q)::EGFP* or *endu-2(E460Q)::EGFP* was able to restore the transcript level 4 h after HS.

Nested effects of time need to be addressed in transcriptomics data. Where are the omics data (transcriptomics and proteomics) deposited? These data do not seem available.

Response: The RNA-seq Raw data have been deposited in BioProject under BioProject ID PRJNA881926. The possessed results of RNA-seq data are located in Supplementary Data. 1-3. The raw data of Proteomics is upload ProteomeXchange under the project accession PXD041872. And the processed proteomics data is available in Supplementary Data. 5.

Detailed feedback:

There are too many language errors scattered throughout the text, and also several overstatements or (sub)sentences with unclear relevance used. E.g. line 4 “Any” organism is “constantly” exposed to ... , or lines 53-54 which refer to research on mice, but are phrased as if human-based. Please don't focus on these two examples only during revision; there are several more sentences in need of

attention in the text; especially the introduction is sensitive to this, but also elsewhere. I would advise to scrutinize the text so that phrasing may be nuanced, language errors corrected, and the reader will be able to focus on what matters: the findings and interpretations. As a minor comment on the side: the authors regularly indicate whether they believe certain topics are researched more/less; this is unnecessary and I would propose to slim the text down to the scientific narrative only.

Response: We admit that writing error free English is a real challenge for us as non-native speaker. With help of the reviewers and editorial board we will try our best to eliminate these small typos and grammatical errors. We also have tried to rephrase our text for more focusing on the scientific significant of the work.

Lines 34-46 Are these orthologs, or simply examples of ENDOU family members? How are the examples given (viral, human proteins) relevant for the topic and ENDOU-2 focus in this work? *C. elegans* seems to have two ENDOU proteins, I would find it more informative to use the introduction to help the reader understand their relation to other ENDOU family members and to heat (and cold – unless there are good reasons to assume that is mechanistically different here) stress, than to see these examples.

Response: The purpose of this description is to provide a comprehensive summary of the ENDOU family, which currently has limited knowledge, with most of the information coming from the virus field. Despite their diverse origins, these ENDOU proteins share similar molecular functions, as indicated by the conserved motifs and amino acid residues responsible for RNA binding and cleavage. However, given the scarcity of information on the biological roles of other ENDOU proteins, it remains unclear whether they function similarly to worm ENDOU-2. To date, no study has described the functions of ENDOU-1, another worm ENDOU protein. We have modified the text and hope that this summary will provide a better understanding of the ENDOU family and serve as a starting point for future research.

Line 45 If ENDOU-2 is a secreted protein, does it have cell-autonomous actions? Apart from the author's previous work, do we know whether any of the functional implications are due to cell-(non)-autonomous actions? If both exist, could the introduction clarify this?

Response: Yes, all three available studies on ENDOU-2 in *C. elegans* have suggested its cell-autonomous function. We have updated the introduction to reflect this clarification.

Line 61 The logic/narrative of the current introduction appears a bit random. Even when revisiting it after having read everything, it still did not fit smoothly with what I summarized out of the entire work: "This work tests the hypothesis that *C. elegans* endu-2 is specifically involved in regulation of gene expression after (but not during) heat shock, and explores some of the potential mechanistic ways it might achieve this."

The authors mention a broader goal: to understand (assumedly via an example study on ENDOU-2?) how RBPs may affect transcription through co-transcriptional RNA processing (cf lines 30-33?), but that is not exactly what is being tested in this work. If this is the overall goal, then why ENDOU-2 and no other RBP? Why both acute and hormetic treatments, how is that relevant for that conceptual question? ... Or is the goal to understand differences between acute HS response and long-term hormetic effects in an ectothermic context? If so, then why the focus on ENDOU-2? Also, in this scenario, starting from a current view on hormetic lifespan effects and known regulatory mechanisms may be interesting? In any case, I am a bit confused by the motivation for the experiments done here, which the authors can probably easily address by adjustments to their introduction.

Response: We appreciate the question and criticism raised by reviewer #1. Our project was motivated by the question of whether ENDU-2 might have a specific function in the soma, given that we were able to detect its expression in somatic cells. Our goal was to gain a better understanding of ENDU-2 protein through further characterization. In the revised version, we have reframed our introduction to better reflect our motivation for this study.

It is stated that strains were maintained at 20°C, however, the authors previously reported that at this temperature, there is progressive loss of brood size over generations in *endu-2* loss of function mutants, ultimately leading to sterility (temperature-dependent Mrt phenotype). Aren't the *endu-2* mutants cultured at 15, then, which is when they do not have any such problems? When were they changed to 20 °C? If this happened, was this also done for all other experimental strains, so that there are no temperature confounders in the observations? How many generations prior to executing experiments on them?... wait, this is probably 'explained' in lines 625-626. So the question would be: are the first five generations fine? This paper should stand sufficiently on its own, so I would appreciate a more extensive description in the methods here, rather than an off-hand reference to previous work, for something that is quite relevant to the current study. For example, how did they deal with the tissue-specific rescue strains, then? Are we dealing with other temperature histories of different strains? After how many generations is an epigenetic effect of temperature negligible? Please explain.

Response: In general, unless otherwise specified, we maintained all strains at 15°C and used several adult hermaphrodites for egg-laying at 20°C. Animals used for experiments were grown at 20°C from embryonic stage. One reason for this breeding strategy is to eliminate the potential influence of the germline Mrt phenotype, as *endu-2(lf)* animals do not exhibit this phenotype at 15°C. We have added this information about culturing to the Methods section.

Furthermore, our results indicate that *endu-2* expression in neurons can preserve germline immortality but has no function in heat hormesis, whereas *endu-2* expression in muscles cannot protect germline function but is capable of mediate the beneficial effects of HS. Therefore, any germline problems in our rescue strains are unlikely to interfere with the heat hormesis response.

The authors should specify what "day 1 of adulthood" precisely is, and how they generated synchronized populations. Nearly all experiments rely on (begin with) day 1 adults, but without defining this, the manuscript does not meet the reporting requirements.

Response: We selected mid L4 animals and performed HS 18 h later. We have added this information to the method section.

Fig 1 & lines 90-96: there are a number of non-trivial considerations to be made with these lifespan experiments. For one, it is unclear how the LogRank p values shown for panels b-i (all survival curves) were derived, and how this analysis took replicate experiments into account. Some of the p values seem odd; for example, for panel h, 0.5953 is a weirdly low p value, when looking at the survival curves. This could be due to showing pooled data, while properly recognizing independent experiments in the statistical analysis, but it needs to be sorted out, because it is impossible to assess at the moment. Survival data are variable (e.g. already easily visible in this figure if one imagines all control conditions that are presumedly effectless in the same graph); that is by no means the authors' doing, but a well-recognized reality of the readout. The authors however are faced with combining this reality with only small (if any) effects observed in panels e-i; therefore showing the replicate experiments (e.g. in supplement), rather than only pooled data, as well as clarifying the statistics are a must.

Response: In all three biological replicates for lifespan assays, we simultaneously formed unstressed and stressed lifespan cohorts and observed similar trends in the wild type control, *endu-2* mutant, and rescue strains. However, there were variations in the mean lifespan among different biological replicates. In the revised version, we present a representative lifespan curve for one of the replicates but summarize all the replicates in Supplementary Figures 1 and 2. Statistical analysis of lifespan was carried out with Log-rank (Mantel-Cox) test provided Graphpad Prism 9.3.1 (471).

In addition, one very worrying fact that is left unaddressed, is that the endogenous rescue does not fully rescue. This may mean the effect seen in panel b is actually not due to *endu-2*, but rather due to a background mutation or an epistatic effect. It may alternatively mean that the construct they used for rescue, a fusion protein with GFP, is unable to do so. What are the results with a proper endogenous rescue (e.g. using a trans-splicing sequence to avoid the GFP being attached to ENDU-2, or even omitting it altogether)? If these are different, then also all tissue-specific rescues will need to be redone. If the results are similar to those of panel e, then what to make of this conceptually, in comparison with the effect observed in panel b? When testing the endogenous rescue, I would advise to also include control conditions in the same experiments and graphs – how much of the wt vs *endu-2(tm4977)* effect seen in these strains is really due to *endu-2*?

Then, is it possible to consider effect sizes also in the tissue specificity? For example, the muscle rescue looks quite similar to the endogenous one. Given this is a secreted protein, perhaps the source of production is less relevant to its ultimate effect on longevity, then the available dose is. Not all tissues may produce it at similar levels, and the tissue-specific promoters used likely do not replicate this faithfully? In fact, conditional knockouts may be more interesting to address the question of tissue-dependent contributions to the phenotype, here?

Response: We appreciate the concern raised by reviewer #1 and have performed the following experiments to address it:

1. We tested another *endu-2* loss-of-function allele, *by188*, which has a small deletion in Exon 1, resulting in an early stop codon. This allele is supposed to be a strong loss-of-function if not a null mutant. We found that *endu-2(by188)* animals exhibited the same Mrt phenotype as *endu-2(tm4977)²* and were also defective in lifespan extension upon short HS (Supplementary Fig. 1c).

2. To investigate whether the EGFP fusion might impair ENDU-2 function in our rescue strains, we tested our *endu-2::EGFP* CRISPR knock-in strain, *endu-2(by190[endu-2::EGFP])*. We found that these animals displayed about 25% lifespan extension upon hormetic heat stress (Supplementary Fig. 1e), suggesting that the EGFP fusion to *endu-2* is unlikely to abrogate its activity with respect to heat hormesis.

3. As our rescue strain expresses much more ENDU-2::EGFP than the EGFP CRISPR knock-in strains, we tested whether excessive overexpression might be the cause of the moderate rescue effect of lifespan. For this, we tested additional rescue lines with less strong *endu-2::EGFP* overexpression (Supplementary Fig. 2c, d). We found that these two rescue lines exhibited a more significant rescue of lifespan, indicating that the beneficial effect of ENDU-2 on lifespan upon hormetic HS is only significant without excessive overexpression of this protein.

4. Finally, we performed one additional assay to measure the beneficial effects of hormetic HS: improved survival upon a subsequent HS. We found 1 hour of HS at the adult stage on day one benefited wild-type but not *endu-2(lf)* mutant animals (Fig. 1e). Notably, our *endu-2* rescue transgenic line, which only showed a moderate lifespan extension upon hormetic HS, completely

rescued the thermotolerance defect in mutant animals. Together with the observations above, we suggest that although ENDU-2 has a beneficial effect in response to hormetic HS, its excessive overexpression might be detrimental in aged animals and could titrate away its beneficial effect on lifespan.

Lines 95-96: Related to the above: it is stated ENDU-2 is a secreted protein. Would it not be equally likely that these tissues are not the main ones “requiring” it, but happen to be the main ones producing it, depending on the strength of the promoters used?

Response: Although ENDU-2 is secreted, the source of its production affects its activity. For its non-cell autonomous activity in the germline, only neuronal and intestinal expressed ENDU-2 can preserve germline immortality, while muscular and somatic gonadal ENDU-2 cannot (Qi et al., 2021). We currently do not know why ENDU-2 from different sources behaves differently. We did observe that ENDU-2 in the germline represses misexpression of somatic-specific genes that are predominantly expressed in neurons and intestine, indicating that ENDU-2 from different tissues may selectively target genes with the same tissue origin. Furthermore, we agree with the reviewers that different tissues may not produce the same amount of proteins. For example, the effect of ENDU-2 in the muscle might be magnified with a strong promoter such as the *myo-3* promoter, as the protein level of ENDU-2 in the muscle is relatively weak compared to that in the intestine. We have added this point in the Results section. To further investigate the tissue-dependent contribution of ENDU-2, we attempted to tag ENDU-2 with a degron in 2019 during the revision of our last ENDU-2 manuscript. However, we were unsuccessful in triggering degron-mediated ENDU-2 degradation. We consulted with Abby Dernburg’s lab, and they explained that for proteins enclosed in certain subcellular compartments, such as endosomes or large protein complexes, the ubiquitination-proteasome system may have difficulty degrading degron-tagged protein targets. As a secreted protein, ENDU-2 will largely enter the ER-Golgi and endosomal vesicles, which is supported by our mass spectrometry data of ENDU-2 interactors that includes many factors involved in ER-Golgi transport, exocytosis, and endocytosis. From the localization pattern of ENDU-2 in the intestine, it may exist in condensate-like structures composed of RNA and RNA-binding proteins. An embedded degron that fails to interact with TIR1 may be the reason why auxin-inducible degradation of ENDU-2 did not work.

Line 143 What is the motivation for deciding on these time points (mainly a question for the 4h one, obviously)? This may also be a good place to already insert the notion elaborated further on in lines 580-586, that this is a dynamic process and temporal choices have to be made to study its dynamics from an omics perspective.

Response: First, we aimed to examine the transcriptome after a recovery period; however, an extended recovery period was not feasible as the *endu-2(lf)* mutant exhibits a severe egg-laying defect resulting from abnormal vulva development^{2, 3}. This defect leads to the death of many animals in days 2 and 3 of adulthood and may significantly influence animal physiology already before death. As a result, we selected a 4-hour recovery period after HS, during which no internal hatching occurs. We have included this clarification in the results section.

Minor: while it is common to use up- and downregulated in omics analyses, the terminology is suggestive in a slightly misleading way. The authors quantified RNA levels, that can be higher/lower or in-/decreased in one condition vs another, but that is a result of more than merely up- or downregulation of transcription. I would suggest to correct this throughout the entire document, or, if that is too cumbersome, to at least very explicitly state what is meant with the terms in the manuscript.

Response: We have modified the text according to the suggestions.

Lines 152-155 (and paragraph below, and later on as well) Yes and no. One concern is that there is not a temporally matched, non-heat shocked control. *C. elegans* transcriptomes are known to be dynamic in early adulthood, so a non-heat shocked population that is 6 hours older than its reference will also display a number of significant transcriptomic differences as a result of that. In the experimental setup used here, these effects are nested with those of the late (incl. hormetic) response. Could the authors explain why they did not include such a control, and adjust their interpretations wherever needed in this manuscript, to take into account that a post-HS animal differs from controls in more ways than only by having undergone the treatment?

Response: The worm used for our RNA-seq experiments were stage matched. We apologize for not explaining it in the last version and now include a diagram illustrating our sample preparation and collection (Supplementary Fig. 3a).

Line 176 what were the selection criteria for these candidates to confirm? How many of these were expected to match expectations, how many were not/true unknowns?

Response: We selected our candidate genes based on two criteria. Firstly, we looked for the most strongly affected genes from each class. Secondly, we prioritized genes with available published primers for RT-qPCR. We have included this information in the revised manuscript. We successfully validated the RNA-seq results for all of the selected genes.

What happens with *endu-2* RNA during and after HS? Would that not be relevant to highlight in the discussion of the results as well?

Response: Our RNA-seq data showed that *endu-2* mRNA level increased slightly (FC=1.76, FDR 9.59E-13) after 1 h of HS, and this elevation was sustained 4 h after HS (FC=1.59, FDR 9.83E-9). Surprisingly, Western blot analysis indicated that the protein level of ENDU-2 decreased after 1 h of HS (Fig. 4f). These findings suggest HS may accelerate protein degradation of ENDU-2 and an increase in *endu-2* mRNA level might serve as a compensatory mechanism possibly through a feedback loop. We have included this information in our revised manuscript.

Lines 181-189 (and more general) ENDU-2 is necessary for hormetic effects after HS (standard lifespan and Cd²⁺ survival alike). This means ENDU-2 needs to “know” that a HS has happened (on and off again). Is ENDU-2 a directly heat sensitive protein (cf also nice observations in lines 521-526)? Can *hsf-1* mutants still raise the *endu-2* response? The latter seems straightforward to test, and would be interesting to strengthen the message of this manuscript? (The same could be asked for *daf-16*, *daf-12* and *swn-1* based on literature given as inspiration in lines 272-273, but *hsf-1* seems most relevant for this question?)

Response: We found that *hsf-1* RNAi did not impede the increased association of ENDU-2 to chromatin during and after HS (Supplementary Fig. 10a, b). This implies that although ENDU-2 appears to be aware of HS and alters its subcellular distribution, it is not governed by HSF-1.

We further investigated whether *swn-1* RNAi affects the chromatin localization of ENDU-2, given that *swn-1* RNAi impaired *irg-2* induction and SWSN-1 was identified as ENDU-2 interactors after HS (Supplementary Fig. 8). *swn-1* RNAi did not impair ENDU-2's chromatin localization but instead seemed to enhance it upon HS (Supplementary Fig. 10c), suggesting that chromatin association of ENDU-2 is independent of SWSN-1. The enhanced association of ENDU-2 with chromatin may be a compensatory response to *swn-1* inhibition. These findings also raise the possibility that ENDU-2

and the SWI/SNF chromatin remodelling complex may interact to regulate the post-HS response, which warrants further investigation. We have included this discussion in the revised manuscript.

Lines 190-194 I agree that is what is suggested, but how to make sense of this? The authors state they previously showed this protein is targeted to the secretory pathway. How do they think it then acts cell-autonomously and/or moves between different subcellular compartments (see also the very nice experiments showing it can do so, but I miss an interpretation that brings together those bits of information)?

Response: We have also questioned ourselves since initiation of this work how a secreted protein could possess a cell-autonomous function? Although we don't have any explanation yet, we do detect ENDU-2 protein in the cells that produce it. We propose that cytoplasmic and nuclear ENDU-2 protein could be exported from the endoplasmic reticulum (ER) through an unknown mechanism or come from an insufficient ER targeting. We have discussed this aspect in the revised manuscript.

Figure 2d can the heat map be provided with all its labels in the extended data? None of the rows are labelled, but this is valuable information. It is impossible to include this in the main figure, but of scientific value to add it to the manuscript.

Response: We added the heat map with the labelling in the Supplementary Data 2.

Extended Data Figure 3

(a) It is unclear to me what the tissue enrichment representation is based on, and the methods section on this (666-669) does not clarify it. What data source(s) feed into this? How many tissues are linked to a transcript? What are the criteria? What does the size of the bars (the “-log₁₀ q value”) mean? I have the impression this was glanced over a bit too superficially, because it matches the expectations. This may remain true after explanation, but as it is, the data shown are not particularly valuable to me.

(b) is there a table containing the detailed information of this available as well? Please refer to where it can be found? Without those data, these are just some numbers in a Venn diagram.

Response: In the last version, we have performed the tissue enrichment analysis with a tool provided by Wormbase (<https://wormbase.org/tools/enrichment/tea/tea.cgi>)⁴. But we understand the concern of the reviewer #1 and reanalysed the tissue enrichment of the ENDU-2 targets. We used the published tissue specific transcriptome data⁵ and compared them with the post-HS responsive genes. We found intestine-only genes were dominant in the ENDU-2 dependent post-HS responsive genes (Supplementary Fig. 5a), suggesting that intestine is probably the most important tissue undergoing post-HS response.

Lines 272-283: how do the authors fit SWSN-1 and ENDU-2 together in their model of heat response/hormesis? This feels incomplete and somewhat out of place in the flow of the manuscript. Why bother testing this if not to connect it to their protein under study? They have an ENDU-2 reporter strain; did they look at its signal after SWSN-1 RNAi? Do we have two proteins affecting *irg-2* transcription, or is one under control of the other?

Response: Our proteomics analysis revealed that ENDU-2 interacts with three factors of SWI/SNF complex: SWSN-1, SWSN-4 and ISW-1, after HS (Supplementary Fig. 8). Moreover, knock-down of components in the SWI/SNF complex strongly impaired *irg-2* activation (Fig. 3b and Supplementary Fig. 7d). While fluorescence microscopy did not provide sufficient evidence to determine whether ENDU-2 affects the association of SWSN-1 with respective chromatin loci or *vice versa*, we did find that the nuclear localization of ENDU-2 and SWSN-1 is independent of each other (Supplementary

Fig. 7f). These observations collectively suggest that ENDU-2 might work together with the SWI/SNF complex to regulate genome organization and thereby impact the post-HS response. This would be a highly interesting topic for future study. Unfortunately, we have not investigated the physical interaction between ENDU-2 and the SWI/SNF complex at current stage, primarily due to the lack of antibodies to detect endogenous SWI/SNF components. We have included these aspects in the revised discussion.

Also, in this paragraph, can the authors please explicitly clarify that RNAi in their paradigm does not target the nervous system? This is relevant in the context of their cell-(non-)autonomous logic that is always at the back of one's mind when reading this manuscript.

Response: We have clarified this in the revised version.

Lines 337-366 are a nice exploration, even though a bit of a lucky shot with then a negative result. Is ZIP-2 under control of SWSN-1? Together with the previous remark, this may tie up the narrative a bit more? Does *C. elegans* have DDIT3 homologs? Might be worth mentioning in this narrative?

Response: Worms lack a homolog of DDIT3. Additionally, we have found that *swsn-1* RNAi does not affect the nuclear localization of ZIP-2::GFP (see the figure below). However, we have decided to remove the ZIP-2 related results from the revised version of the manuscript based on feedback from reviewer #3 as we agree with the reviewer's opinion that including ZIP-2 results in this manuscript may be unnecessary and could potentially distract from the main findings.

Lines 353-360 Transcriptomics does not seem to rely on full-length reads (cf methods). Then why is it permitted to compare read counts in this part of the manuscript?

Response: It is the full length of the *zip-2* transcript. The last thinner line represents the 3'UTR of *zip-2* mRNA. However, we have decided to remove the *zip-2* related data from the manuscript and instead focus more on the role of ENDU-2 in transcription, as suggested by the reviewer #3.

Line 390 and onward and figure 9: this is very nice work, but how to make sense of this for a secretory pathway protein (see above)? Are SWSN-1 or ZIP-2 amongst the proteomic partners? Are there any other interesting partners in the list? Now, it seems, the only thing the MS experiments support, is the lead that ENDU-2 might also exist in the nucleus.

Response: Given our observation of the autonomous functions of ENDU-2, we are also intrigued by the question of how a protein that is designated for the secretory pathway can enter the nucleus and have a cell-autonomous function. We have detected ENDU-2 protein in the somatic cells that produce it, but we do not have a clear understanding of its mechanism of action at present. We speculate that cytoplasmic ENDU-2 may be actively exported out from the ER, or that the ER targeting of ENDU-2 may not be 100% efficient, resulting in some cytosolic translation of ENDU-2 protein. We have included a discussion of this issue in the revised manuscript.

In our MS study, we identified three components of the SWI/SNF complex (SWSN-1, SWSN-4, and ISW-1) as ENDU-2 interactors, but we did not identify any transcription factors. Rather, we found certain histone proteins, three Pol II components and several chromatin-associated proteins as ENDU-2 interactors. In the revised version of the manuscript, we have labelled these ENDU-2-interacting chromatin factors in Supplementary Fig. 8 and Supplementary Data 5.

line 488 provides RPB-2 as “another” hit, but there are no clear hits mentioned before this (unless I missed that?)?

Response: In the revised version, we have provided an expanded description of our MS analysis. In addition to the identification of SWSN-1, SWSN-4, and ISW-1 as ENDU-2 interactors, we have also identified three subunits of RNA polymerase II (RPB-2, RPB-5, and RPB-7). These findings suggest that ENDU-2 may have a direct role in regulating transcription through interaction with Pol II and chromatin factors. We have included a more detailed analysis of the MS data in the revised manuscript.

Lines 439-440 Yes. Or that these amino acids are needed for something else as well? Can that be ruled out or considered unlikely? If so, please mention why?

Response: Our previous RIP-seq and RNA cleavage experiments have shown that E460Q is defective in RNA binding and E454Q possesses even stronger affinity to RNA but defective in RNA cleavage² (Qi et al. 2021). As interaction between Pol II and ENDU-2 is RNA dependent, it is possible that the lack of RNA-binding capacity in E460Q mutant makes it unable to interact with Pol II. For E454Q mutant, the same mutation on the conserved site of Xenopus XendoU abolishes its Mn²⁺ binding ability, which interferes with RNA-cleavage as all ENDOU proteins cleave RNA in a Mn²⁺ dependent manner. Therefore, the loss of Pol II interaction in E454Q could be due to the lack of RNA-cleavage or Mn²⁺ binding activity. However, we cannot exclude the possibility that these amino acids may

have additional functions that are important for their association with Pol II. In the revised version, we have modified our interpretation of the results to avoid overstatement.

Line 490 & extended data 8: AMA-1 was not enriched in the ENDU-2 pulldown MS, why was it chosen for co-IP? Were these the only Pol II proteins that were identified by MS? If so, then please mention, but also, there is no real predictive value in being a “better” hit from the MS analysis? If not, could others please also be indicated in Extended Data Figure 8, and could it be explained why those were not focused on?

Response: We selected AMA-1 and RBP-2 for Co-IP experiments based on the availability of antibodies. Our analysis identified three Pol II components as interactors of ENDU-2, which are highlighted in Supplementary Data 5.

Reviewer #2 (Remarks to the Author):

This paper reports a key role for the endoribonuclease ENDU-2 in transcriptional modulation of the benefits of hormesis. A well-established model of heat shock hormesis is used to examine the mechanisms of ENDU-2. The authors find that ENDU-2 is required for hormetic lifespan extension as well as hormetic resistance to cadmium. The effects of ENDU-2 are reported to require ENDU-2 association with chromatin upon heat shock; recruitment of RNA Pol II; and results in transcription of a distinct set of genes, some of which (“Class III”) occurs after the acute heat shock response has subsided. This post-heat-shock adaptation can be decoupled from the acute heat shock response.

This is a data-rich and appropriately-controlled study in which the (surprising?) conclusion that EDNU-2 acts as a nuclear transcription factor is well-supported. There are a few minor issues to address:

The manuscript does not cite Jia et al, who claimed that the germline effects of EDNU-2 in response to genotoxic stress were mediated largely through CTPS-1 phosphorylation, controlled by down-regulation of PKA and HDA-1. While this may be unrelated to the hormetic effects of EDNU-2, it would be interesting to know if the transcripts of any of these genes were altered in the investigator's transcriptome data.

Reponses: We are aware of the study of Jia et al. and have cited it in the last version (Line 41).

According to our transcriptomic result, *ctps-1* mRNA level is slightly increased upon HS in WT (\log_2 FC = 0.58, FDR = 0.0455) and *endu-2(lf)* (\log_2 FC = 0.80, FDR = 0.0012), 4 h after HS, the expression level was reset back to basal level.

***hda-1* mRNA level was slightly decreased upon HS in WT (\log_2 FC = -0.4, FDR = 0.0262) but not in *endu-2(lf)* (\log_2 FC = -0.1, FDR = 0.0003), 4 h after HS the mRNA level both WT (\log_2 FC = -0.8, FDR = 0.0092) and *endu-2(lf)* (\log_2 FC = -0.6, FDR = 0.0212) animals displayed slightly decreased *hda-1* transcript level.**

lines 439, 440 - The claim that both RNA-binding and cleavage are essential to mediate the beneficial effect of heat hormesis is not the logical conclusion of the data presented. Since the E460Q mutation affects both RNA binding and RNA cleavage, it remains possible that only RNA cleavage is required for the hormetic effect.

Response: We agree with the concern raised by reviewer #2. The lack of interaction between Pol II and the E460Q and E454Q mutants might be due to the loss of RNA cleavage activities, we

acknowledge that these amino acids could also have additional functions that play a role in their association with Pol II. Therefore, we have modified our interpretation of the results in the revised version to avoid overstatement.

The probability values in extended data Fig 5a appear to be switched.

Response: We apologize for our carelessness and have corrected this mistake in the revised version.

Given the proposed role of EDNU-2 as a transcription factor, it would have been informative to have done a ChIP-seq experiment with the EDNU-2::GFP strain, as this could have identified all the genes under EDNU-2 control, and potentially revealed any sequence motif responsible for EDNU-2 chromatin binding. However, given the extensive analysis already presented in this manuscript, this is not something that needs to be added to this study.

Response: We appreciate the positive feedback from the reviewer #2 regarding our EDNU-2 ChIP-qPCR experiment. We acknowledge that ChIP-seq would provide more comprehensive information. Currently we are in the process of conducting EDNU-2 ChIP-seq at various time points and plan to include the results of these experiments in a future follow-up study. Thank you for the suggestion.

Reviewer #3 (Remarks to the Author):

In this manuscript the authors investigate how the benefits of hormetic heat stress are manifested over time on a transcriptional level. This is an interesting, well-executed manuscript that addresses an important question in aging biology and stress-response hormesis. Specific findings and comments under each section heading. EDNU-2 contributes to hormetic heat stress-mediated beneficial effects. The authors show that *endu-2* mutants have no defect in resistance to heat stress but that *endu-2* is required for CdCl₂ resistance and hormetic HS-mediated lifespan extension. The *endu-2* rescue experiments are not convincing and need to be strengthened. Specifically:

- The authors should describe the *endu-2* mutation better and demonstrate or cite previous research confirming loss-of-function.

Response: The *endu-2(tm4977)* mutant harbours a 619 bp deletion that spans part of the *endu-2* promoter, the entire 5' untranslated region to part of the third exon. Our RNA-seq analysis revealed that the *tm4977* allele had less than 10% of reads mapped to exons downstream of the deleted region. Therefore, the *tm4977* allele is likely a null mutant. We have included the information about deletion in the Material and Method section of the revised manuscript.

- The expression pattern of EDNU-2 should be described better and the authors should include a supplementary figure showing the expressing pattern of the EDNU-2::GFP constructs they are using for lifespan.

Response: We have included the description of the expression pattern in the results and added the figures showing the expression pattern of all the tissue specific EDNU-2::EGFP transgenes used in this study (Supplementary Fig. 2).

- It would be interesting to determine whether the benefits of hormetic heat shock extend to other stresses besides CdCl₂ resistance, especially the reported increased resistance to heat stress. This would strengthen their findings that EDNU-2 is involved in conveying multiple hormetic benefits and could be used as additional read-outs in subsequent figures, since the authors only examine CdCl₂ resistance and do not show additional lifespan experiments.

Response: We conducted an additional assay and tested the thermal tolerance of pre-treated animals to a subsequent heat stress. Our results showed that hormetic HS improved the survival of wild-type animals, but not *endu-2(tm4977)* mutants (Fig. 1e). Remarkably, transgenic expression of *endu-2::EGFP* fully restored the thermotolerance phenotype. We also assessed the ability of tissue-specific *endu-2::EGFP* transgene expression to rescue the hormesis defect in *endu-2(lf)* mutants and found that expression in the intestine fully rescued while muscular expression could also has weak rescue. Collectively, these data suggest that ENDU-2 likely functions in the intestine and muscle to mediate the beneficial effects of short HS.

- Lifespan experiments: All lifespan/survival assays should be performed at least three times independently. Typically, lifespan data are not pooled but reported as independent experiments and all data listed in tables. The lifespan data in Figure 1e-i need to be performed with WT controls, to demonstrate that the heat shock induced hormetic benefits. The authors did not include in the methods sections what program log-rank test was calculated with. We recommend OASIS2 (a web-based survival calculation program).

Response: In the previous version, we conducted lifespan experiments with WT control at the same time. To avoid overcrowding in one graph, we split the data into different figures. In the revised version, we have included data from three biological replicates and presented one representative replicate in the main figure. We have also summarized lifespan curves of all biological replicates in Supplementary Fig. 1 and Fig. 2. Statistical analysis for lifespan was analysed using Log-rank (Mantel-Cox) test provided by Graphpad Prism 9.3.1 (471), which is a widely used statistic method for lifespan.

-In Fig 1e to 1i, the levels of ENDU-2 rescue are not clear and need to be addressed. Was the *endu-2* level back to WT? Or they are overexpressing *endu-2*?

Response: In our first submission we used BR7295, which has strongly overexpressed *ENDU-2::EGFP* in comparison to our *EGFP* CRISPR knock-in strain by 190. To know whether excessive overexpression of *ENDU-2* is responsible for the insufficient rescue of the mutant lifespan, we generated two additional *endu-2::EGFP* rescue lines with a slight overexpression. These two rescue lines were able to rescue the *endu-2(lf)* mutant lifespan phenotype upon hormetic HS. Notably, the BR7295 with strong *endu-2* overexpression completely restored thermotolerance in *endu-2(lf)* animals (Fig. 1e). These data suggest that although *ENDU-2* mediates the beneficial effect of hormetic HS, too much *ENDU-2* might have detrimental consequences later in life, eventually negating the earlier benefits acquired.

-Lifespan assays were performed by adding FUDR. Even only they used for the first 7 days, the FUDR still did affect animals. The authors should at least show one set of lifespan without adding FUDR.

Response: Lifespan experiments without FUDR were performed at the very beginning of the project using a protocol adapted from Kumsta et al. (2017)⁶, which subjected animals to a 45-minute 36°C heat shock. We observed a significant extension of lifespan in wild-type animals, but not in *endu-2(lf)* mutants (Supplementary Fig. 1d). However, we later decided to use a 35°C heat shock for 1 hour, as this condition is more commonly used in the literature. Furthermore, we introduced FUDR treatment due to the high percentage (60-80%) of bagging observed in *endu-2(lf)* animals. We have now included the lifespan data obtained without FUDR treatment in the Supplementary Fig. 1d.

- Fig 1a should use two-way AVOVA not multiple t test for statistics.

Response: We replaced multiple t test with the two-way ANOVA in the revised version.

Homeostatic heat stress results in an *ENDU-2* dependent reprogramming of the transcriptome after HS. In this section the authors nicely demonstrate the *endu-2* dependent gene transcription changes that occur immediately after heat shock and compare them to transcriptional changes occurring after the acute heat shock with 4h recovery. We have some suggestions to improve the clarity of this section.

- Line 184-185, more information about the function of these genes chosen as representative for each class would be helpful, especially in the case of *irg-2*, which is further explored.

Response: According to the literature, HSP-16.2 is a chaperone that facilitates protein folding. *zip-10* and *pqm-1* encode transcription factors that have been implicated in immune responses^{7, 8}. Although the molecular function of IRG-2 protein is unknown, its transcript level increases in response to pathogen infection⁹, suggesting a potential role in innate immunity response. We selected these genes for qPCR analysis because their transcripts exhibited the most significant increase following heat stress, and published primer sets are available for their quantification. We have included an explanation for our selection criteria in the revised version of the manuscript.

- Line 190-191, "...genes was enriched in the intestine and muscle (Supplementary Fig 3a)." The authors should explain this better, are they known intestinal and muscle-specific genes? This is important to clarify since the others did bulk RNASeq and not did not isolate tissues.

Response: In the last version, we have performed the tissue enrichment analysis with a tool provided by Wormbase (<https://wormbase.org/tools/enrichment/tea/tea.cgi>)⁴. We now used the published tissue specific transcriptome data⁵ and compared them with the post-HS responsive genes. Our analysis revealed that the post-HS responsive genes are enriched in transcripts specifically expressed in the intestine (Supplementary Fig. 5a), strongly suggesting that the intestine is the primary tissue responsible for mediating the beneficial effects of heat hormesis.

- Figure 2d would benefit from a schematic of the classes as presented in Fig 7, or show the pattern of the selected genes representative for each class over time.

Response: We have added a schema according to the suggestion.

- In Fig 2f, it seems that all selected genes are increased in *endu-2(lf)* at 20°C. What are the statistics of these genes for WT and *endu-2(lf)* at 20 °C? The authors should include a comparison of their RNAseq data of WT with *endu-2* mutants under basal conditions. How many genes are differentially regulated in the mutants and do these genes fall into the HS-regulated classes? The authors should use two-way ANOVA for their statistical tests.

Response: We have compared the post-HS responsive genes with those that are up-regulated in *endu-2* at 20°C (Supplementary Fig. 3b) in the last version. Since about 40% of the post-HS responsive genes are negatively affected by *ENDU-2* at 20°C, *ENDU-2* can both activate and inhibit the same gene in a cellular context dependent manner. Why *ENDU-2* shows opposite regulatory effects towards the same genes under different conditions is currently unknown. RNA-Seq data are now shown in the Supplementary Data. 1 and 3. We replaced one-way ANOVA with the two-way ANOVA in the revised version and also made a comparison for selected genes at basal condition. According to two-way ANOVA, expression level of the selected genes in WT and *endu-2(lf)* animals are not statistically different, although *endu-2(lf)* animals show at least several folds increase of the transcript level. This is because HS induced fold change in WT animals was much stronger than that between the two strain at 20°C and two-way ANOVA analysis considers the two independent variables (stress and genotype).

- Extended Data Fig2: The authors need to include an additional repeat and should include either Lifespans or another hormetic benefit in these analyses. Why would these specific genes be required for CdCl₂ resistance? What is the physiological function and relevance here?

Response: We have included the third replicates of Cd²⁺ assay in the revised version. In addition, we tested whether these genes are involved in hormesis induced thermotolerance (Supplementary Fig. 4e)

-ENDU-2 positively regulates gene expression at the transcriptional level after HS. The authors postulate that ENDU-2, a mRNA-binding endoribonuclease regulates gene expression on a transcriptional level and they are using transcriptional reporter constructs to differentiate between transcriptional and post-transcriptional regulation and compare Class I and Class II genes. These experiments do not answer this question.

- Since Fluorescence readouts require the synthesis of a fluorescent protein and are thus not direct read-outs for transcription.- The experimental set-up for class I and Class II fluorescent reporters is different. TJ375 worms will be fluorescent after a 1h HS and 2h Recovery, but not after just 1h HS which was reported in their RNASeq experiment, most likely due to a delay in protein synthesis of gfp. To determine the different transcriptional class the reporter strains need to be imaged at the same HS and recovery times. Moreover in line 264 the authors state the *irg-2* reporter was measured after 8h while the figure legend states 16h.

Response: The TJ375 strain, which carries the *Is[hsp-16.2p::GFP]* transgene, was subjected to heat shock (HS) for 4 hours prior to microscopy (as described in the figure legend), without any recovery period. For imaging of the *Is[irg-2p::mCherry]* reporter, all images presented in the main figure were taken after 16 hour recovery from a 1 h HS. This overnight recovery period was chosen to better align with the lab's working schedule, as opposed to an 8-10-hour recovery period. In Supplementary Fig. 6, we compared animals exposed to continuous HS with those that had recovered for different periods from 1 hour of HS. We show that activation of the *irg-2* transcriptional reporter is detectable after 8 hours of recovery but not under continuous exposure to HS, suggesting that the *irg-2* transcriptional reporter is inducible in the recovery period after HS rather than prolonged HS.

- In line 267, the authors state "Surprisingly, hormetic HS failed to activate *irg-2p::mCherry*". Why is this surprising, they already showed this by RNASeq and qRTPCR?

Response: The *irg-2* reporter is a transcriptional reporter that measures the activity of the *irg-2* promoter. Since ENDU-2 is known to be an mRNA-binding protein that can affect RNA stability, it may also impact the stability of *irg-2* mRNA. If this is the case, ENDU-2 would not affect the activity of the *irg-2* transcriptional reporter. Therefore, our results suggesting ENDU-2 as an activator of transcriptional initiation is an unexpected surprising result if we take its only known role in regulating mRNA stability into consideration.

- In extended Data Fig 5A, the p-values seem to be swapped.

Response: We apologize for our carelessness and have corrected this mistake in the revised version.

-Activation of the bZIP transcriptional factor ZIP-2 in the post-HS phase is independent of ENDU-2. The authors hypothesized that ZIP-2 (responsible for regulating *irg-2* gene expression) is activated by ENDU-2 and that ZIP-2 would in turn activate the post-HS gene expression, they find a possible activation of ZIP-2 which is however independent of ENDU-2. - This section seems unnecessary and distracts from the main finding of this manuscript. Should the authors wish to keep this section in the

manuscript they should address whether ZIP-2 is required for the hormetic benefits similarly to *irg-2*.
- The images of nuclear localization are fuzzy. The authors need to provide better images if they tried to claim ZIP-2 translocating into nucleus after HS but not in *endu-2(lf)*.

Response: We agree with the opinion of the reviewer #3 and have removed all ZIP-2 related data from the manuscript.

-ENDU-2 localizes in the nucleus and associates with chromatin
In this section the authors identified that ENDU-2 is localized in the nucleus and cytoplasmic fraction and can be immunoprecipitated with Histone H3. Nuclear localization of ENDU-2 is increased with HS. This section could be strengthened with additional experiments:- The ENDU-2 IP followed by Mass spec is not well described in the results or in the methods sections and should be improved. The authors should consider describing the nuclear proteins that ENDU-2 is associated with in more details. Was Histone H3 found in mass spec? Maybe their own data can help strengthen the claim that ENDU-2 is associated with chromatin

Response: In the revised version, we have provided a comprehensive list of all ENDU-2-associated proteins that are either known or predicted to be localized to the nucleus. This information can be found in Supplementary Fig. 8 and Data 5.

- In line 405-406 the RNAase treatment and RNA-independence should be described in more detail as it is not clear. Did the authors have any control for RNase A digestion in Fig 4c? How did they know all the RNA in their reaction was gone?

Response: We have added a more detailed description of the RNA dependent protein interaction and include a control experiment for RNase A digestion in Supplementary Fig. 12.

- Since the authors are capable to isolate nuclear and cytoplasmic fraction, does HS and HS followed by recovery induce more ENDU-2 in nuclear fraction?

Response: We have investigated ENDU-2::EGFP protein levels in nuclear and cytoplasmic fractions under various conditions (as shown in Fig. 4f, g). We observed a slight reduction in ENDU-2 protein levels in the cytoplasm following HS, but not in the nucleus. These findings suggest that the increased chromatin association of ENDU-2 following HS is not due to an increase in nuclear import, but rather a result of elevated enrichment in chromatin association.

-Fig 4b and 4d, they should use the chromatin marker (e.g. histone) to examine if ENDU-2 colocalized with chromatin, rather than DAPI staining.

Response: We have co-stained ENDU-2::EGFP with GFP antibody (monoclonal mouse), histone H3 with anti H3 antibody (monoclonal rabbit) and tried three different secondary antibodies Alexa Fluor 488, 555, and Cy5. However, except 555, the other two secondary antibodies produced strong background that interfered with H3 or GFP signal. Therefore, we could not detect ENDU-2::EGFP and H3 at the same time. With 555 secondary antibody, H3 signal completely overlapped with signal from DAPI stained DNA (Supplementary Fig. 9), suggesting that colocalization of ENDU-2::EGFP with DAPI signal is a strong hint for its chromatin association.

- Both RNA-binding and -cleavage activities of ENDU-2 are essential for mediating heat hormesis. The authors use RNA-binding and RNA-cleaving mutants of ENDU-2 to determine their function in heat hormesis and find that while both activities are required for the transcriptional regulation of *irg-2* upon HS. The imaging data is confusing and does not fit with their nuclear cytoplasmic fractionation, which again should be performed +/- HS.

- Line 441/442, if the authors want to identify whether ENDU-2(E454Q) and E460Q are chromatin associated, they should provide histone H3 IP results and use chromatin marker in extended fig 7 as well. The current data they provided only shows ENDU-2(E454Q) and E460Q exist in nucleus but not chromatin associated

Response: We conducted histone H3 immunoprecipitation experiments and were able to detect co-immunoprecipitation of ENDU-2(E454Q) and ENDU-2(E460Q), indicating their association with chromatin (Supplementary Fig. 11b). Furthermore, the imaging data presented in Supplementary Fig. 11c support the chromatin association of ENDU-2(E460Q)::EGFP, which forms a condensate-like granule structure similar to ENDU-2(wt)::EGFP that is enhanced upon HS (Supplementary Fig. 11e). These findings suggest that the RNA-binding activity of ENDU-2 may not be essential for chromatin association. For ENDU-2(E454Q) we observed protein localization in the nucleus but without formation of condensate (Supplementary Fig. 11c). However, we cannot exclude the possibility that the E454Q or E460Q mutation may interfere with other properties of the protein besides RNA-binding or -cleavage, and thus we have modified our interpretation of these observations in the results.

We were unable to perform colocalization experiments between ENDU-2(E454Q) or (E460Q) with H3 due to limitations with our available secondary antibodies, as previously mentioned.

- Line 447, "...incapable to localize on chromatin..." and line 448, "...the chromatin association of ENDU-2..." again, their data did not support these statements.

Response: Based on our H3 IP result, E454Q probably associates with chromatin. However, it is diffused distributed in the nucleus cannot form condensate-like structure which is observed by ENDU-2(wt). We have modified the text based on the new data.

- An additional readout for hormetic benefits, such as LS or stress resistance should be added

Response: We have include the thermotolerance assay and found that neither transgenic *endu-2(E454Q)::EGFP* nor *endu-2(E460Q)::EGFP* is able to improve thermotolerance of animals after pre-treatment with 1 h HS (Fig. 5c).

ENDU-2 activates RNA polymerase II to trigger post HS response Here the authors provide intriguing and nice mechanistic evidence of how ENDU-2 affects the transcriptional response after hormetic HS. 2. In fig 6a and 6d, what are the major bands (a bit smaller than 75 kDa) in the anti-GFP blot when IP AMA-1? They are abundant and even appear in GFP control group.

Response: These bands are from the antibody (IgG) heavy chain. We have labelled them in the revised figure.

& Minor comments:1. Why do authors include right Y axis in all survival curve figures? Is it really necessary?

Response: No. It is unnecessary. We have removed it.

2. Line 41, stresse should be "stresses."

3. The label of the promoter of the gene in transgenes is incorrect. According to Caenorhabditis nomenclature 4.5.4.

(http://www.wormbook.org/chapters/www_nomenclature/caenoromenclature.html#sec4.5), the lowercase p is used to indicate the promoter of the gene rather than uppercase.

4. In line 113, a p is missing in *myo-2p::mCherry* transgene. It should be *myo-2p::mCherry*.

5. Line 118, mediating “the” beneficial effect of hormetic HS.
6. Line 320, y is missing in *irg-2p::mCherry*.
7. Line 380, a space is lacking between “two independent...”
8. Figure 4b “intestine (20 oC)” e is missing.
9. Merge images for fig 4d are also helpful for readers to see the colocalization even though the authors provide colocalization coefficient.
10. The authors should quantify the colocalization coefficient for extended fig 7c and 7d as well and provide merge images of chromatin marker and ENDU-2.
11. Line 440, to mediate “the” beneficial effect.
12. Line 877, ref 36 “*Caenorhabditis elegans*” is missing after “from” in the title.

Response: We appreciate your feedback regarding our mistakes in nomenclature and spelling, and we have made the corrections in the revised version of our manuscript. Thank you for bringing this to our attention.

Reference

1. Jovic, K. *et al.* Temporal dynamics of gene expression in heat-stressed *Caenorhabditis elegans*. *PLoS One* **12**, e0189445 (2017).
2. Qi, W. *et al.* The secreted endoribonuclease ENDU-2 from the soma protects germline immortality in *C. elegans*. *Nat Commun* **12**, 1262 (2021).
3. Jia, F., Chi, C. & Han, M. Regulation of Nucleotide Metabolism and Germline Proliferation in Response to Nucleotide Imbalance and Genotoxic Stresses by EndoU Nuclease. *Cell Rep* **30**, 1848-1861.e1845 (2020).
4. Angeles-Albores, D., N Lee, R.Y., Chan, J. & Sternberg, P.W. Tissue enrichment analysis for *C. elegans* genomics. *BMC Bioinformatics* **17**, 366 (2016).
5. Kaletsky, R. *et al.* Transcriptome analysis of adult *Caenorhabditis elegans* cells reveals tissue-specific gene and isoform expression. *PLoS Genet* **14**, e1007559 (2018).
6. Kumsta, C., Chang, J.T., Schmalz, J. & Hansen, M. Hormetic heat stress and HSF-1 induce autophagy to improve survival and proteostasis in *C. elegans*. *Nat Commun* **8**, 14337 (2017).
7. Irfan Afridi, M. *et al.* The bZIP transcription factor BATF3/ZIP-10 suppresses innate immunity by attenuating PMK-1/p38 signaling. *Int Immunol* **35**, 181-196 (2023).
8. Rajan, M. *et al.* NHR-14 loss of function couples intestinal iron uptake with innate immunity in. *Elife* **8** (2019).
9. Estes, K.A., Dunbar, T.L., Powell, J.R., Ausubel, F.M. & Troemel, E.R. bZIP transcription factor zip-2 mediates an early response to *Pseudomonas aeruginosa* infection in *Caenorhabditis elegans*. *Proc Natl Acad Sci U S A* **107**, 2153-2158 (2010).

REVIEWERS' COMMENTS

Reviewer #1 (Remarks to the Author):

The authors revised their manuscript with great diligence, and addressed the vast majority of the concerns. I fully recommend public communication of this work.

One minor remark: I still somewhat disagree with the logic of lines 370-373 that this is "the logic in the field". There is an enormous body of literature focusing on experience-dependent plasticity. Perhaps the authors do not feel this is their field, but the concept surely is relevant to the idea mentioned here, and it would be nice to acknowledge that or nuance the phrasing.

Some typos:

17 hormetic

62 HDA-1

160 Based

188 significant

199 the hsp16.1 gene encodes ... // chaperone

201 considered to participate

205 ...zip-10 or irg-2, or pqm-1 mutation ...

221-2 How these two opposite functions are achieved ...

228 upstream of the irg-2 start codon to ...

230 under reference growth conditions, ...

249 levels

275 unstressed conditions

277 the antibody stain against histone H3 completely overlapped with the DAPI ...

286-7 or the SWI/SNF complex could be involved

292 or the endu-2 gene

302 different from wild-type ENDU-2 regarding their

303 Transgenically expressed

306 similar to

307 mRNA levels

322 the interaction of ENDU-2 with multiple

326 co-IP

331 displayed a chromatin localization similar to

368-9 In the wild, organisms frequently encounter transient stress experiences.

403 out of the ER

406 in the cytoplasm

410 to reach the cytoplasm

411 by the signal peptide recognition particle

412-3 the ENDU-2 peptide across the ER membrane, eventually resulting in cytoplasmic retention of the ENDU-2 protein.

413 A mechanistic understanding of the distribution of ENDU-2

414-5 cell-autonomous and -non-autonomous

417 have so far predominantly been linked

424 and Pol II

426 activated a reporter gene

448 putative

449 DNA segments are expected to form

Reviewer #2 (Remarks to the Author):

The authors have addressed my concerns about the original submission, and appear to

have comprehensively addressed the (more extensive) issues raised by my fellow reviewers.

Reviewer #3 (Remarks to the Author):

I am satisfied with the responses and changes to the manuscript, but would recommend detailed proof reading by the authors and editorial staff.

Author Response:

We are happy to see that all three reviewers were satisfied with the responses and changes to the manuscript. We are especially grateful that the reviewer #1 had pointed out these typos in our text and have corrected all of them. In addition, we have replaced the phrasing in the discussion lines 370-373 with the following text:

Hormesis refers to a unique response characterized by a biphasic reaction to increased quantities of substances or stimuli that are typically harmful at high doses but exhibit beneficial effects within a specific low-dose range known as the hormetic zone. The prevailing theory that supports the advantageous outcomes is the overcompensation of defensive responses. Consequently, extensive research in hormesis has primarily concentrated on elucidating how low doses of toxins or stressors can activate various stress responses. One classical illustration of hormesis is the induction of gene expression of heat-shock proteins and autophagy regulators by hormetic HS, resulting in lifespan extension and improved stress resistance in *C. elegans*.

Reviewer #1 (Remarks to the Author):

The authors revised their manuscript with great diligence, and addressed the vast majority of the concerns. I fully recommend public communication of this work.

One minor remark: I still somewhat disagree with the logic of lines 370-373 that this is "the logic in the field". There is an enormous body of literature focusing on experience-dependent plasticity. Perhaps the authors do not feel this is their field, but the concept surely is relevant to the idea mentioned here, and it would be nice to acknowledge that or nuance the phrasing.

Some typos:

17 hormetic

62 HDA-1

160 Based

188 significant

199 the hsp16.1 gene encodes ... // chaperone

201 considered to participate

205 ...zip-10 or irg-2, or pqm-1 mutation ...

221-2 How these two opposite functions are achieved ...

228 upstream of the irg-2 start codon to ...

230 under reference growth conditions, ...

249 levels

275 unstressed conditions

277 the antibody stain against histone H3 completely overlapped with the DAPI ...

286-7 or the SWI/SNF complex could be involved

292 or the endu-2 gene

302 different from wild-type ENDU-2 regarding their

303 Transgenically expressed

306 similar to
307 mRNA levels
322 the interaction of ENDU-2 with multiple
326 co-IP
331 displayed a chromatin localization similar to
368-9 In the wild, organisms frequently encounter transient stress experiences.
403 out of the ER
406 in the cytoplasm
410 to reach the cytoplasm
411 by the signal peptide recognition particle
412-3 the ENDU-2 peptide across the ER membrane, eventually resulting in cytoplasmic retention of the ENDU-2 protein.
413 A mechanistic understanding of the distribution of ENDU-2
414-5 cell-autonomous and -non-autonomous
417 have so far predominantly been linked
424 and Pol II
426 activated a reporter gene
448 putative
449 DNA segments are expected to form

Reviewer #2 (Remarks to the Author):

The authors have addressed my concerns about the original submission, and appear to have comprehensively addressed the (more extensive) issues raised by my fellow reviewers.

Reviewer #3 (Remarks to the Author):

I am satisfied with the responses and changes to the manuscript, but would recommend detailed proof reading by the authors and editorial staff.